# Layer-wise Gradient Disentanglement: Decoupling Semantics and Preferences in Direct Preference Optimization

Mengyang Li [1]   Shuang Liu [1]   Zhong Zhang [1]

## Abstract

Direct Preference Optimization (DPO) has become the dominant approach for aligning large language models with human preferences. However, standard DPO treats all preference pairs uniformly, overlooking the heterogeneous nature of the learning problem: some samples demand sophisticated semantic understanding of the prompt, while others require nuanced discrimination between similar responses. We argue that these two objectives should be disentangled during training. Through gradient analysis, we identify a layer-wise localization phenomenon where semantic complexity predominantly drives lower-layer updates while preference uncertainty modulates upper layers. Building on this insight, we propose Gradient-Guided Disentangled DPO (GDO-DPO), a curriculum framework that independently regulates learning pace along each dimension based on layer-specific gradient stability. Experiments on UltraFeedback and HH-RLHF demonstrate consistent improvements, with GDO-DPO outperforming DPO by 4.1% on AlpacaEval 2.0 and showing particularly strong gains on reasoning-intensive tasks.

## 1. Introduction

Direct Preference Optimization (DPO) (Rafailov et al., 2023) has emerged as a practical alternative to the multi-stage RLHF pipeline (Christiano et al., 2017; Ouyang et al., 2022), reformulating the reinforcement learning objective into a classification loss that can be optimized directly. This simplification has catalyzed rapid development: IPO (Azar et al., 2024), KTO (Ethayarajh et al., 2024), SimPO (Meng et al., 2024), ORPO (Hong et al., 2024), CPO (Xu et al., 2024), and SPPO (Wu et al., 2025c) have further refined the optimization target. Yet a fundamental question remains underexplored: *are all preference pairs created equal?*

Consider two examples from a typical preference dataset. The first prompt asks a model to debug recursive code with memoization. This task demands parsing nested logic and verifying algorithmic correctness, while the preference signal is unambiguous since the code either works or fails. The second prompt requests a short poem about autumn. Semantic understanding is straightforward, but distinguishing preferred from rejected responses depends on subtle stylistic judgments where reasonable annotators may disagree. These tasks impose fundamentally different challenges on orthogonal dimensions, yet standard DPO processes them identically.

We posit that preference optimization conflates two distinct learning objectives. The first is **representation learning**: constructing internal representations sufficient to comprehend the input context $x$. The second is **discrimination learning**: refining decision boundaries to distinguish preferred response $y_w$ from rejected response $y_l$. Interpretability studies support this functional separation. Lower Transformer layers have been shown to encode syntax and local semantics (Clark et al., 2019), middle layers store factual knowledge (Geva et al., 2021; Meng et al., 2022), and upper layers perform task-specific reasoning (Voita et al., 2019). If these functions are localized, the gradients induced by different sample characteristics should also exhibit spatial structure.

This paper provides the first analysis of how sample difficulty distributes across network depth during DPO training. We introduce two complementary metrics: *semantic complexity* $\mathcal{C}_{\text{sem}}$, measuring the representational demand imposed by the prompt, and *preference uncertainty* $\mathcal{U}_{\text{pref}}$, quantifying the ambiguity between response pairs. Through gradient analysis on a simplified linear model, we derive that these dimensions induce separable effects: semantic complexity predominantly scales gradients in representation layers (lower and middle), while preference uncertainty modulates gradients in discrimination layers (upper). Empirical verification on Llama-3-8B and Mistral-7B confirms this

[1]Tianjin Key Laboratory of Wireless Mobile Communications and Power Transmission, Tianjin Normal University, Tianjin, China, 300387. Correspondence to: Zhong Zhang <zhong.zhang8848@gmail.com>.

*Proceedings of the 43rd International Conference on Machine Learning*, Seoul, South Korea. PMLR 306, 2026. Copyright 2026 by the author(s).

*layer-wise gradient localization* phenomenon, with high-$\mathcal{C}_{\text{sem}}$ samples concentrating gradients in layers 0–16 and high-$\mathcal{U}_{\text{pref}}$ samples activating layers 20–32.

Building on these findings, we identify a failure mode we term *gradient interference*. When samples with high preference ambiguity are presented before representations stabilize, the large gradients arising from discrimination errors propagate throughout the network. In representation layers, these gradients reflect preference noise rather than representational deficiency, potentially distorting the world model that the pretrained LLM has constructed. This interference can manifest as hallucination or superficial alignment that fails to generalize.

To address this problem, we propose **Gradient-Guided Disentangled DPO (GDO-DPO)**, a curriculum learning framework that monitors layer-specific gradient statistics to independently regulate representation and discrimination learning. Unlike conventional curricula that rely on scalar difficulty metrics such as loss or perplexity (Bengio et al., 2009), GDO-DPO maintains two pace parameters controlling the active training set along each difficulty axis. The algorithm advances semantic complexity when representation-layer gradients stabilize, and increases preference uncertainty when discrimination accuracy on clear examples reaches a threshold. This state-aware progression ensures that the model consolidates prompt understanding before tackling ambiguous preferences, respecting the functional hierarchy of Transformer layers. Our contributions are as follows:

- We provide the first layer-wise analysis of DPO training dynamics, revealing that semantic complexity and preference uncertainty induce separable gradient patterns across network depth, with consistent results across model architectures.

- We propose GDO-DPO, a closed-loop curriculum algorithm that leverages internal gradient states to disentangle representation and discrimination learning, requiring minimal computational overhead.

- Extensive experiments on two datasets (UltraFeedback and HH-RLHF) and two model families demonstrate state-of-the-art performance, with particularly large gains on reasoning-intensive tasks (+0.50 on MT-Bench Math category).

**Conflict of Interest Disclosure.** The authors have no financial conflicts of interest to disclose.

## 2. Related Work

**Preference Alignment.** Aligning LLMs with human values originated with RLHF (Christiano et al., 2017; Ouyang et al., 2022), which trains a reward model and optimizes the policy via PPO (Schulman et al., 2017). DPO (Rafailov et al., 2023) eliminates explicit reward modeling by deriving a closed-form loss from the optimal policy. Subsequent refinements address various limitations: IPO (Azar et al., 2024) adds regularization to prevent overfitting; KTO (Ethayarajh et al., 2024) enables training without paired preferences; SimPO (Meng et al., 2024) introduces length-normalized margins; ORPO (Hong et al., 2024) integrates preference learning into supervised fine-tuning. More recent work tackles robustness and exploration: Dr. DPO (Wu et al., 2025b) applies distributional robustness to handle noisy preferences, $\alpha$-DPO (Wu et al., 2025a) adapts reward margins dynamically, and COPO (Bai et al., 2025) incorporates count-based exploration for online RLHF. These methods focus on the optimization objective while treating data consumption as uniform. Li et al. (2026) propose a meta-learning paradigm that fuses intrinsic feedback signals during preference alignment, highlighting the importance of adaptive training strategies. GDO-DPO addresses a complementary dimension: how sample presentation order affects learning, and can combine with any of these loss functions.

**Curriculum Learning.** The principle that presenting samples from easy to hard improves generalization dates back to Bengio et al. (2009). Self-paced learning (Kumar et al., 2010) allows models to select samples based on current competence. In NLP, curricula have leveraged sentence length (Platanios et al., 2019), perplexity (Xu et al., 2020), and domain specificity. Recent applications to LLM training include loss-based curricula for instruction tuning (Lee et al., 2024) and self-paced strategies for RLHF (Wu et al., 2025c). However, these approaches rely on scalar difficulty metrics that conflate multiple difficulty sources. Li & Zhao (2026) incorporate difficulty awareness into learning curve extrapolation, demonstrating benefits of structured difficulty modeling. GDO-DPO uses layer-specific gradient statistics, enabling finer-grained control that respects Transformer functional hierarchy.

**Training Dynamics and Interpretability.** Studies have established that Transformer layers exhibit functional specialization: early layers process syntax (Clark et al., 2019), middle layers store factual knowledge (Geva et al., 2021; Meng et al., 2022), and later layers perform task-specific reasoning (Voita et al., 2019). Recent theoretical work analyzes gradient dynamics in attention mechanisms (Yang et al., 2024) and characterizes how different components converge at varying rates (Im & Li, 2025). Razin et al. (2025) identify likelihood displacement as a failure mode in DPO, while Bansal et al. (2023) show that lower-layer representation quality impacts downstream performance. Kumar et al. (2022) demonstrate that fine-tuning can distort pretrained features and hurt out-of-distribution performance,

providing motivation for representation preservation during alignment. Ren et al. (2023) study how task head preparation affects backbone features during fine-tuning. Ren & Sutherland (2025) propose a framework for analyzing LLM fine-tuning dynamics, identifying a "squeezing effect" closely related to our gradient interference phenomenon. Our work bridges interpretability and optimization by translating layer-wise gradient localization into a practical training algorithm. Li et al. (2025a) provide a complementary perspective by analyzing training dynamics in image classification, revealing how sample difficulty interacts with model learning stages.

## 3. Theoretical Analysis

This section develops the theoretical foundation for our approach. We define two dimensions of sample difficulty, analyze gradient flow in a tractable model, and verify the predicted patterns empirically.

### 3.1. Preliminaries

Consider a preference dataset $\mathcal{D} = \{(x_i, y_i^w, y_i^l)\}_{i=1}^N$, where $x$ denotes the prompt, $y^w$ the preferred response, and $y^l$ the rejected response. The DPO objective is:

$$\mathcal{L}_{\text{DPO}}(\theta) = -\mathbb{E}_{\mathcal{D}}\left[\log \sigma\left(\beta \cdot \Delta r_\theta(x, y^w, y^l)\right)\right], \quad (1)$$

where $\Delta r_\theta = \log \frac{\pi_\theta(y^w|x)}{\pi_{\text{ref}}(y^w|x)} - \log \frac{\pi_\theta(y^l|x)}{\pi_{\text{ref}}(y^l|x)}$ is the implicit reward margin. The scalar loss aggregates contributions from all samples without distinguishing their underlying difficulty characteristics.

### 3.2. Two Dimensions of Sample Difficulty

We decompose sample difficulty into two components that capture distinct aspects of the learning problem.

**Definition 3.1** (Semantic Complexity). For a prompt $x$, the semantic complexity $\mathcal{C}_{\text{sem}}(x)$ quantifies the representational demand imposed on the model. We estimate the predictive entropy via Monte Carlo sampling:

$$\mathcal{C}_{\text{sem}}(x) \approx -\frac{1}{K}\sum_{k=1}^K \log \pi_{\text{ref}}(y_k|x), \quad (2)$$

where $\{y_k\}_{k=1}^K$ are responses sampled from $\pi_{\text{ref}}(\cdot|x)$.

High semantic complexity indicates that the model exhibits high uncertainty when generating responses, suggesting that the prompt requires sophisticated understanding. Examples include multi-step reasoning problems, technical explanations, and tasks requiring domain expertise.

**Definition 3.2** (Preference Uncertainty). For a response pair $(y^w, y^l)$ given prompt $x$, the preference uncertainty $\mathcal{U}_{\text{pref}}$ quantifies the ambiguity of the preference signal:

$$\mathcal{U}_{\text{pref}}(y^w, y^l|x) = \exp\left(-\left|r^*(y^w|x) - r^*(y^l|x)\right|\right), \quad (3)$$

where $r^*$ denotes the ground-truth reward, approximated via reward model scores in practice.

High preference uncertainty arises when the reward gap between responses is small, making discrimination difficult. This occurs in subjective tasks such as creative writing or style-dependent instructions.

*Remark* 3.3. The semantic complexity measure $\mathcal{C}_{\text{sem}}$ depends on the reference model rather than intrinsic linguistic properties. This is a deliberate design choice: our goal is to quantify the difficulty that the *current model* faces when processing a prompt, not abstract complexity. Empirically, we find that $\mathcal{C}_{\text{sem}}$ correlates strongly with model-agnostic measures such as prompt perplexity under a separate language model (Spearman $\rho > 0.75$), suggesting that model-dependent and model-independent notions of complexity are aligned in practice. Details are provided in Appendix E.

### 3.3. Gradient Analysis in a Linear Model

To develop theoretical intuition, we analyze a simplified linear model. Consider an encoder $W_E \in \mathbb{R}^{d \times d_{\text{in}}}$ representing lower layers and a discrimination head $w_H \in \mathbb{R}^d$ representing upper layers, with implicit reward $r_\theta(x, y) = w_H^\top W_E \phi(x, y)$. Let $\Delta\phi = \phi(x, y^w) - \phi(x, y^l)$ denote the feature difference between response pairs.

**Proposition 3.4** (Layer-Wise Gradient Sensitivity). *Under the linear model, the gradient norms satisfy:*

$$\|\nabla_{w_H}\mathcal{L}\| = \beta\alpha \cdot \|W_E \Delta\phi\|, \quad (4)$$
$$\|\nabla_{W_E}\mathcal{L}\|_F = \beta\alpha \cdot \|w_H\| \cdot \|\Delta\phi\|, \quad (5)$$

*where $\alpha = 1 - \sigma(\beta\Delta r_\theta) \in (0, 1)$ is the margin sensitivity factor.*

The proof is provided in Appendix B. The key insight is that the encoder gradient scales directly with $\|\Delta\phi\|$, the feature-space discriminability of response pairs, while both gradients are modulated by $\alpha$, which captures preference ambiguity. We hypothesize that $\|\Delta\phi\|$ correlates with semantic complexity $\mathcal{C}_{\text{sem}}$: prompts requiring richer representations induce more distinguishable response features. Empirical validation (Appendix E) confirms this connection ($\rho = 0.58$). When preferences are clear (large reward margin), $\alpha$ is small and gradients are suppressed. When preferences are ambiguous (small margin), $\alpha$ approaches 0.5 and gradients are amplified.

This analysis reveals gradient sensitivity coupling: gradients are jointly determined by input complexity and margin sensitivity. While the simplified linear model exhibits coupled magnitude scaling, the *spatial localization* observed

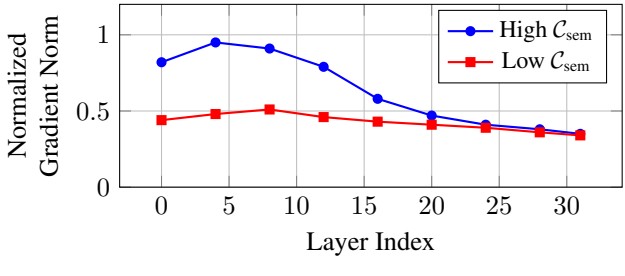

*(a) Stratified by semantic complexity*

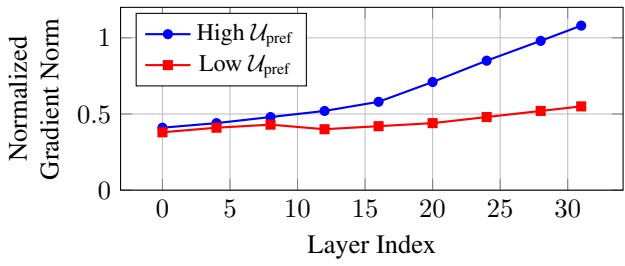

*(b) Stratified by preference uncertainty*

*Figure 1.* **Layer-wise gradient localization in Llama-3-8B.** Gradient norms averaged over 1000 samples from top/bottom difficulty terciles. High semantic complexity concentrates gradients in lower layers (approx. 0–16), while high preference uncertainty amplifies upper-layer gradients. Note that our algorithm uses a conservative boundary $L_{mid} = 21$ to preserve factual knowledge stored in middle layers. Both separations are statistically significant ($p < 0.01$).

in Figure 1 arises from the architectural inductive bias of deep Transformers, where $W_E$ (conceptually lower layers) constructs the features $\phi$ while $w_H$ (upper layers) performs discrimination. Our linear analysis formally characterizes the driving forces ($\|\Delta\phi\|$ vs. $\alpha$) that activate these respective components.

**Gradient Interference.** When samples with high $\mathcal{U}_{pref}$ are presented before representations stabilize, the large $\alpha$ amplifies gradients throughout the network. In representation layers, these amplified gradients reflect preference ambiguity rather than representational deficiency, potentially destabilizing features that were converging toward useful representations.

**Definition 3.5** (Gradient Interference). Let $g_{rep}^{(t)}$ denote representation layer gradients at step $t$. Gradient interference occurs when $\mathrm{Var}(g_{rep}^{(t)})$ increases due to high-$\mathcal{U}_{pref}$ samples while $\mathcal{C}_{sem}$-induced gradients have not yet stabilized.

### 3.4. Empirical Verification

We verify the theoretical predictions on Llama-3-8B using UltraFeedback. Following prior work on Transformer interpretability (Clark et al., 2019; Geva et al., 2021), we partition layers into functional groups: lower layers (0–10) primarily encoding syntax and local semantics, middle layers (11–21) storing factual knowledge, and upper layers (22–31) performing task-specific reasoning. For gradient analysis, we aggregate lower and middle layers as the representation component and upper layers as the discrimination component. Figure 1 presents the layer-wise gradient norms stratified by difficulty dimensions. We compute the L2 norm of gradients at each layer, averaged over 1000 samples from each difficulty tercile (top and bottom 33%) and normalized by the maximum across all layers. The results confirm our theoretical predictions. High-$\mathcal{C}_{sem}$ samples induce elevated gradients in layers 0–16, with the effect diminishing in upper layers. Conversely, high-$\mathcal{U}_{pref}$ samples show relatively flat gradients in lower layers but substantial amplification in layers 20–31. This separation validates the two-dimensional

decomposition and motivates layer-aware curriculum design.

Beyond gradient magnitude, we also examine gradient *directions* via cosine similarity within difficulty groups. In layers 0–16, high-$\mathcal{C}_{sem}$ samples exhibit cosine similarity of 0.82, indicating coherent gradient directions, while high-$\mathcal{U}_{pref}$ samples show only 0.41. In layers 20–31, the pattern reverses (0.38 vs. 0.79). This directional coherence confirms that the localization reflects structured, aligned updates rather than random magnitude variation (see Appendix H for details).

### 3.5. Generalization Beyond the Linear Model

The linear model analysis provides intuition but relies on simplifying assumptions. Real Transformers include residual connections, attention mechanisms, and layer normalization that could potentially mix gradients across layers. We conduct additional experiments to assess whether the gradient localization phenomenon generalizes. We repeat the gradient analysis on Mistral-7B-v0.3, which has a different architecture from Llama-3. Figure 7 in Appendix D shows qualitatively similar patterns: high-$\mathcal{C}_{sem}$ samples concentrate gradients in lower layers while high-$\mathcal{U}_{pref}$ samples activate upper layers. The correlation between layer-wise gradient profiles across the two models exceeds 0.85 for both difficulty dimensions, suggesting that gradient localization is architecture-agnostic. To assess whether residual connections blur the gradient separation, we compute the correlation between adjacent layer gradients. If residual paths dominated, we would expect high correlation throughout the network. Instead, correlation drops from $\rho > 0.85$ in layers 0–18 to $\rho < 0.65$ in layers 19–31 (see Figure 8 in Appendix D), indicating that the functional separation persists despite residual connections.

## 4. Methodology

Building on the theoretical insight that semantic complexity and preference uncertainty induce separable gradient patterns, we propose Gradient-Guided Disentangled DPO

(GDO-DPO). This section details the algorithm design and monitoring mechanisms.

## 4.1. Design Principles

GDO-DPO is built on three principles derived from our analysis. First, **separation of learning phases**: the curriculum maintains independent progress along the complexity and uncertainty axes, allowing the model to consolidate prompt understanding before tackling ambiguous preferences. Second, **state-aware progression**: rather than following a predetermined schedule, the curriculum monitors internal gradient statistics to determine when the model is ready for more challenging samples. Third, **minimal overhead**: the monitoring mechanisms leverage quantities computed during standard backpropagation, ensuring practical applicability without significant additional cost.

## 4.2. Bi-Dimensional Curriculum Space

**Difficulty Quantification.**    For each sample $(x_i, y_i^w, y_i^l) \in \mathcal{D}$, we precompute $\mathcal{C}_{\text{sem}}(x_i)$ via Equation 2 by sampling $K = 8$ responses from the reference policy, and $\mathcal{U}_{\text{pref}}(y_i^w, y_i^l | x_i)$ via Equation 3 using reward scores from the dataset.

We convert absolute values to normalized ranks $R_{\text{sem}}(x_i) \in [0, 1]$ and $R_{\text{unc}}(x_i, y_i^w, y_i^l) \in [0, 1]$, representing percentile positions within the dataset. Rank normalization ensures robustness to outliers and enables consistent threshold interpretation across datasets.

**Active Set Construction.**    At training step $t$, the algorithm maintains two pace parameters $\lambda_{\text{sem}}^{(t)}, \lambda_{\text{unc}}^{(t)} \in [0, 1]$ that define the active training set:

$$\mathcal{D}_t = \Big\{ (x, y^w, y^l) \in \mathcal{D} : R_{\text{sem}}(x) \leq \lambda_{\text{sem}}^{(t)} \\ \land R_{\text{unc}}(x, y^w, y^l) \leq \lambda_{\text{unc}}^{(t)} \Big\}. \quad (6)$$

This formulation enables traversal of the complexity-uncertainty plane along paths determined by the model's internal state, rather than being constrained to a fixed diagonal as in scalar difficulty curricula. Figure 2 illustrates the bi-dimensional curriculum space.

## 4.3. Layer-Aware Monitoring

The key innovation of GDO-DPO is using internal gradient statistics to regulate curriculum progression. We define two complementary monitors.

**Representation Stability Monitor.**    This metric determines readiness for increased semantic complexity. Based on Proposition 3.4, high complexity manifests as elevated gradients in representation layers. We monitor the relative

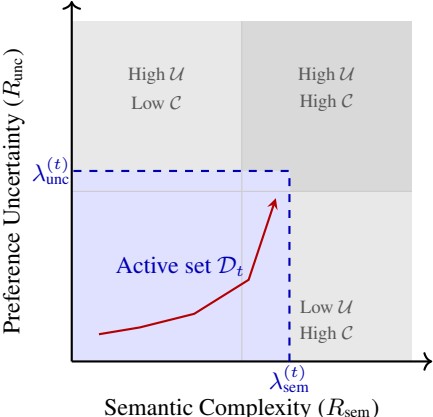

*Figure 2.* **Bi-dimensional curriculum space.** The active training set $\mathcal{D}_t$ (shaded blue) is bounded by pace parameters $\lambda_{\text{sem}}^{(t)}$ and $\lambda_{\text{unc}}^{(t)}$. Darker regions indicate higher overall difficulty. The trajectory illustrates how the boundary expands during training.

gradient energy:

$$S_{\text{rep}}^{(t)} = \text{EMA}_\gamma \left( \frac{\sum_{\ell \in \mathcal{L}_{\text{rep}}} \|\nabla_{\theta_\ell} \mathcal{L}_t\|^2}{\sum_{\ell \in \mathcal{L}_{\text{disc}}} \|\nabla_{\theta_\ell} \mathcal{L}_t\|^2 + \epsilon} \right), \quad (7)$$

where $\mathcal{L}_{\text{rep}} = \{0, \dots, L_{\text{mid}}\}$ and $\mathcal{L}_{\text{disc}} = \{L_{\text{mid}} + 1, \dots, L\}$ partition the layers, $\text{EMA}_\gamma$ denotes exponential moving average with decay $\gamma$, and $\epsilon$ prevents numerical instability. We set $L_{\text{mid}} = \lfloor 2L/3 \rfloor$ based on interpretability findings (Meng et al., 2022); sensitivity analysis in Section 5 confirms robustness to this choice.

A high $S_{\text{rep}}$ value indicates that gradient mass concentrates in representation layers, suggesting active feature adjustment. As representations stabilize, $S_{\text{rep}}$ decreases. We advance $\lambda_{\text{sem}}$ when $S_{\text{rep}}^{(t)} < \tau_{\text{stable}}$.

**Discrimination Readiness Monitor.**    This metric determines readiness for increased preference uncertainty. We evaluate discrimination accuracy (computed via log-probabilities) on a held-out validation set consisting of samples within the current complexity boundary:

$$A_{\text{disc}}^{(t)} = \mathbb{E}_{(x, y^w, y^l) \sim \mathcal{D}_{\text{val}}^{(\lambda_{\text{sem}}^{(t)})}} \big[ \mathbb{1} \left( \log \pi_\theta(y^w | x) > \log \pi_\theta(y^l | x) \right) \big], \quad (8)$$

where $\mathcal{D}_{\text{val}}^{(\lambda)}$ contains validation samples with $R_{\text{sem}} \leq \lambda$ and low $R_{\text{unc}}$ (bottom 30%).

If the model cannot reliably distinguish clear preferences at the current complexity level, introducing ambiguous pairs provides noisy signal. We increase $\lambda_{\text{unc}}$ when $A_{\text{disc}}^{(t)} > \tau_{\text{acc}}$.

## 4.4. Curriculum Dynamics

The interplay between the two monitors creates an emergent learning pattern. Early in training, $S_{\text{rep}}$ is high as the model

adapts to the data distribution. The algorithm keeps $\lambda_{\text{unc}}$ low, focusing on semantically simple prompts with clear preferences. This allows representation layers to stabilize without interference from ambiguous discrimination signals.

Once $S_{\text{rep}}$ drops below threshold, $\lambda_{\text{sem}}$ increases to introduce more complex prompts. Simultaneously, as $A_{\text{disc}}$ improves, $\lambda_{\text{unc}}$ begins to increase, exposing the model to finer preference distinctions. Both boundaries eventually expand until the full dataset is active.

### 4.5. Pace Update Rules

We employ adaptive step sizes that accelerate when the model shows consistent readiness:

$$\Delta_{\text{sem}}^{(t+1)} = \Delta_{\text{sem}}^{(t)} \cdot \begin{cases} 1.1 & \text{if } S_{\text{rep}}^{(t)} < 0.8 \cdot \tau_{\text{stable}} \\ 1.0 & \text{otherwise} \end{cases}, \quad (9)$$

$$\Delta_{\text{unc}}^{(t+1)} = \Delta_{\text{unc}}^{(t)} \cdot \begin{cases} 1.1 & \text{if } A_{\text{disc}}^{(t)} > 1.1 \cdot \tau_{\text{acc}} \\ 1.0 & \text{otherwise} \end{cases}. \quad (10)$$

The acceleration factor and margin thresholds are hyperparameters; we find the algorithm robust to reasonable choices (see Section 5).

GDO-DPO modifies only data presentation order, not the loss function, and can be combined with any DPO variant as demonstrated in Section 5. Algorithm 1 summarizes the complete procedure.

## 5. Experiments

We evaluate GDO-DPO on standard alignment benchmarks, analyze training dynamics, and conduct ablation studies to validate design choices.

### 5.1. Experimental Setup

**Models.** We use Llama-3-8B-Instruct and Mistral-7B-v0.3 as base models, representing current open-source LLMs with strong instruction-following capabilities.

**Datasets.** We train on two preference datasets: UltraFeedback (Cui et al., 2024), containing approximately 60,000 binarized preference pairs across diverse instruction types, and HH-RLHF (Bai et al., 2022), containing approximately 170,000 pairs focused on helpfulness and harmlessness. The two datasets differ substantially in domain distribution, enabling assessment of generalization.

**Baselines.** We compare against the base SFT model and six preference optimization methods: DPO (Rafailov et al., 2023), IPO (Azar et al., 2024), SimPO (Meng et al., 2024), ORPO (Hong et al., 2024), CPO (Xu et al., 2024), and SPPO (Wu et al., 2025c). We also include two curriculum

---

**Algorithm 1** GDO-DPO

1: **Input:** Policy $\pi_\theta$, reference $\pi_{\text{ref}}$, dataset $\mathcal{D}$ with precomputed ranks $(R_{\text{sem}}, R_{\text{unc}})$
2: **Hyperparameters:** $\tau_{\text{stable}}, \tau_{\text{acc}}, \Delta_{\text{sem}}, \Delta_{\text{unc}}, \gamma, L_{\text{mid}}, E_{\text{eval}}$
3: Initialize $\lambda_{\text{sem}}, \lambda_{\text{unc}} \leftarrow \Delta_{\text{sem}}, \Delta_{\text{unc}}$; $\quad S_{\text{rep}} \leftarrow 1.0$
4: **for** each training step $t$ **do**
5: $\quad \mathcal{D}_t \leftarrow \{(x, y^w, y^l) \in \mathcal{D} : R_{\text{sem}}(x) \leq \lambda_{\text{sem}} \wedge R_{\text{unc}}(x) \leq \lambda_{\text{unc}}\}$
6: $\quad$ Sample minibatch $B \subset \mathcal{D}_t$; compute loss $\mathcal{L}$ and gradients $\nabla_\theta \mathcal{L}$
7: $\quad G_{\text{rep}} \leftarrow \sum_{\ell=0}^{L_{\text{mid}}} \|\nabla_{\theta_\ell} \mathcal{L}\|^2; \quad G_{\text{disc}} \leftarrow \sum_{\ell=L_{\text{mid}}+1}^{L} \|\nabla_{\theta_\ell} \mathcal{L}\|^2$
8: $\quad S_{\text{rep}} \leftarrow \gamma \cdot S_{\text{rep}} + (1 - \gamma) \cdot G_{\text{rep}}/(G_{\text{disc}} + \epsilon)$
9: $\quad$ Update $\theta$ via optimizer
10: $\quad$ **if** $t \mod E_{\text{eval}} = 0$ **then**
11: $\quad\quad$ Compute $A_{\text{disc}}$ on validation set
12: $\quad\quad$ **if** $S_{\text{rep}} < \tau_{\text{stable}}$ **then**
13: $\quad\quad\quad \lambda_{\text{sem}} \leftarrow \min(1, \lambda_{\text{sem}} + \Delta_{\text{sem}})$
14: $\quad\quad\quad$ Update $\Delta_{\text{sem}}$ via Eq. 9
15: $\quad\quad$ **end if**
16: $\quad\quad$ **if** $A_{\text{disc}} > \tau_{\text{acc}}$ **then**
17: $\quad\quad\quad \lambda_{\text{unc}} \leftarrow \min(1, \lambda_{\text{unc}} + \Delta_{\text{unc}})$
18: $\quad\quad\quad$ Update $\Delta_{\text{unc}}$ via Eq. 10
19: $\quad\quad$ **end if**
20: $\quad$ **end if**
21: **end for**
22: **Return** $\pi_\theta$

---

baselines: Curriculum-Loss, which sorts samples by initial DPO loss, and Curriculum-PPL, which sorts by prompt perplexity.

**Evaluation.** We report results on MT-Bench (Zheng et al., 2023), a multi-turn dialogue benchmark scored by GPT-4 on a 1–10 scale; AlpacaEval 2.0 (Li et al., 2023), measuring length-controlled win rate against GPT-4 Turbo; and Arena-Hard (Li et al., 2025b), containing challenging prompts requiring nuanced reasoning.

**Implementation.** We implement GDO-DPO using the TRL library with AdamW optimizer, learning rate $5 \times 10^{-7}$, batch size 128, cosine schedule, and 3% warmup. For GDO-DPO, we set $\tau_{\text{stable}} = 1.2$, $\tau_{\text{acc}} = 0.65$, $\Delta_{\text{sem}} = \Delta_{\text{unc}} = 0.1$, $\gamma = 0.9$, and $L_{\text{mid}} = 21$ for both Llama-3-8B and Mistral-7B (32 layers). We report mean and standard deviation over 3 runs.

### 5.2. Main Results

Table 1 presents the main comparison on UltraFeedback. GDO-DPO achieves consistent improvements across all benchmarks and models. On Llama-3-8B, GDO-DPO outperforms the strongest baseline (SimPO) by +2.7% on Al-

*Table 1.* **Main Results on UltraFeedback.** Performance across alignment benchmarks. We report mean $\pm$ std over 3 runs. Best results in **bold**, second-best underlined.

| Method | Llama-3-8B-Instruct | | | Mistral-7B-v0.3 | | |
|---|---|---|---|---|---|---|
| | MT-Bench | AlpacaEval 2.0 | Arena-Hard | MT-Bench | AlpacaEval 2.0 | Arena-Hard |
| SFT | 8.05±0.04 | 22.1±0.5% | 14.2±0.4% | 7.65±0.05 | 18.5±0.6% | 12.8±0.5% |
| DPO | 8.22±0.05 | 28.4±0.8% | 18.5±0.6% | 7.92±0.04 | 24.1±0.7% | 16.3±0.5% |
| IPO | 8.18±0.04 | 27.9±0.7% | 18.1±0.5% | 7.88±0.05 | 23.5±0.6% | 16.0±0.4% |
| SimPO | 8.25±0.03 | 29.8±0.6% | 19.2±0.5% | 8.01±0.04 | 25.8±0.5% | 17.5±0.4% |
| ORPO | 8.20±0.04 | 28.1±0.7% | 18.3±0.5% | 7.95±0.04 | 24.5±0.6% | 16.5±0.5% |
| CPO | 8.23±0.04 | 29.2±0.6% | 18.8±0.5% | 7.98±0.04 | 25.2±0.5% | 17.1±0.4% |
| SPPO | 8.24±0.04 | 29.5±0.7% | 19.0±0.5% | 7.99±0.04 | 25.5±0.6% | 17.3±0.4% |
| Curr-Loss | 8.26±0.04 | 29.1±0.7% | 18.8±0.5% | 7.95±0.05 | 24.6±0.6% | 16.8±0.5% |
| Curr-PPL | 8.24±0.05 | 28.9±0.8% | 19.0±0.6% | 7.98±0.04 | 24.8±0.5% | 16.5±0.4% |
| **GDO-DPO** | **8.41±0.03** | **32.5±0.6%** | **21.4±0.4%** | **8.15±0.03** | **27.4±0.5%** | **19.1±0.4%** |

*Table 2.* **Results on HH-RLHF (Llama-3-8B).**

| Method | MT-Bench | AlpacaEval 2.0 | Arena-Hard |
|---|---|---|---|
| DPO | 8.18±0.04 | 27.2±0.7% | 17.8±0.5% |
| SimPO | 8.21±0.04 | 28.5±0.6% | 18.4±0.5% |
| **GDO-DPO** | **8.35±0.03** | **31.2±0.5%** | **20.5±0.4%** |

*Table 3.* **Compatibility with IPO and ORPO (Llama-3-8B).**

| Method | Arena-Hard | AlpacaEval 2.0 |
|---|---|---|
| IPO | 18.1% | 27.9% |
| GDO-IPO | 20.2% (+2.1) | 30.1% (+2.2) |
| ORPO | 18.3% | 28.1% |
| GDO-ORPO | 20.0% (+1.7) | 29.8% (+1.7) |

*Table 4.* **Combination with SimPO (Llama-3-8B).**

| Method | MT-Bench | AlpacaEval 2.0 | Arena-Hard |
|---|---|---|---|
| DPO | 8.22±0.05 | 28.4±0.8% | 18.5±0.6% |
| SimPO | 8.25±0.03 | 29.8±0.6% | 19.2±0.5% |
| GDO-DPO | 8.41±0.03 | 32.5±0.6% | 21.4±0.4% |
| **GDO-SimPO** | **8.48±0.03** | **33.1±0.5%** | **22.0±0.4%** |

pacaEval 2.0 and +2.2% on Arena-Hard. The gains over standard DPO are larger: +4.1% and +2.9% respectively. Similar patterns hold for Mistral-7B, with improvements of +1.6% over SimPO on both benchmarks.

The improvement on Arena-Hard, which contains complex reasoning queries, exceeds that on AlpacaEval in relative terms. This supports our hypothesis that disentangling representation and discrimination learning benefits demanding prompts where stable representations are essential. Curriculum-Loss and Curriculum-PPL provide modest improvements over random sampling but fall short of GDO-DPO, confirming that scalar difficulty metrics conflate the two dimensions and fail to respect layer-wise structure.

**Generalization to HH-RLHF.** Table 2 shows results when training on HH-RLHF, which has different domain characteristics from UltraFeedback. GDO-DPO maintains consistent improvements across all metrics, demonstrating that the approach generalizes across dataset distributions. The gains are comparable to those observed on UltraFeedback, suggesting that the layer-wise gradient localization phenomenon is not dataset-specific.

**Combination with SimPO.** To verify that GDO-DPO addresses an orthogonal dimension from loss function improvements, we apply the curriculum strategy to SimPO. As shown in Table 4, GDO-SimPO achieves 33.1% on AlpacaEval 2.0 and 22.0% on Arena-Hard, improving over both GDO-DPO and SimPO individually. This confirms that data ordering and loss design provide complementary

benefits, and that GDO can serve as a general enhancement applicable to various preference optimization methods.

**Broad Compatibility with DPO Variants.** Since GDO modifies only data presentation order, it is agnostic to the loss function. Table 3 confirms this on IPO and ORPO: GDO-IPO improves Arena-Hard by +2.1% and GDO-ORPO by +1.7%, comparable to GDO-DPO (+2.9%) and GDO-SimPO (+2.8%). Full results are provided in Appendix F.

### 5.3. Category-Level Analysis

Figure 3 presents MT-Bench scores broken down by category. The performance gap between GDO-DPO and DPO varies systematically with task type. Categories requiring complex reasoning show substantial gains: Math (+0.50), Reasoning (+0.46), and Coding (+0.47). In contrast, stylistic categories show smaller improvements: Writing (+0.07) and Roleplay (+0.06).

This pattern provides direct evidence for our framework.

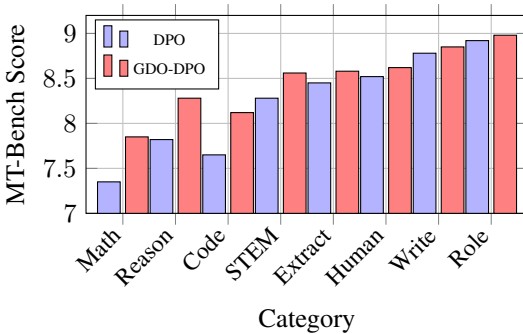

*Figure 3.* **MT-Bench Score by Category (Llama-3-8B).** GDO-DPO shows largest gains on reasoning-intensive categories (Math, Reasoning, Coding) where semantic complexity is high.

Reasoning tasks impose high semantic complexity because they require the model to parse logical structures, track dependencies, and verify correctness. These tasks benefit from stabilized representation learning that GDO-DPO provides. Stylistic tasks, on the other hand, have low semantic complexity but potentially high preference uncertainty, as quality judgments depend on subjective criteria. For these tasks, the discrimination component matters more than representation, explaining the smaller gains.

### 5.4. Training Dynamics Analysis

We analyze how GDO-DPO's curriculum evolves during training and its effect on model internals.

**Curriculum Progression.** Figure 4 shows the evolution of pace parameters $\lambda_{\text{sem}}$ and $\lambda_{\text{unc}}$ alongside the monitoring metrics $S_{\text{rep}}$ and $A_{\text{disc}}$. The algorithm automatically discovers a two-phase structure without explicit programming. In the first phase (steps 0–400), $\lambda_{\text{sem}}$ increases rapidly as $S_{\text{rep}}$ drops, indicating that representation layers stabilize on progressively complex prompts. During this phase, $\lambda_{\text{unc}}$ remains low, shielding the model from ambiguous preferences. In the second phase (steps 400–900), with representations stable, $\lambda_{\text{unc}}$ accelerates as discrimination accuracy $A_{\text{disc}}$ improves. Both boundaries eventually reach 1.0, at which point the full dataset becomes active.

**Gradient Interference Analysis.** To directly test the gradient interference hypothesis, we compare gradient variance in representation layers under different training regimes. Figure 5 shows three conditions: standard DPO with random sampling, a high-$\mathcal{U}_{\text{pref}}$-first curriculum that prioritizes ambiguous samples, and GDO-DPO. The high-$\mathcal{U}_{\text{pref}}$-first curriculum causes elevated gradient variance in early training (steps 0–200), confirming that premature exposure to ambiguous preferences destabilizes representation learning. This curriculum also achieves lower final performance (Arena-Hard: 16.2% vs 18.5% for DPO), providing causal evidence that gradient interference harms outcomes. Note

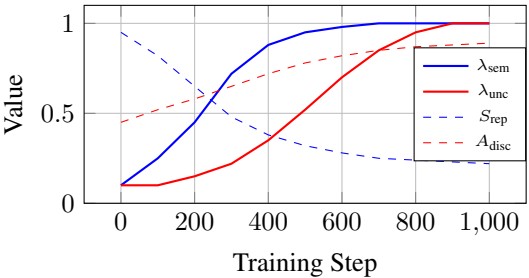

*Figure 4.* **Curriculum Dynamics During Training.** Solid lines show pace parameters; dashed lines show monitoring metrics. The semantic pace $\lambda_{\text{sem}}$ increases first as $S_{\text{rep}}$ stabilizes, followed by the uncertainty pace $\lambda_{\text{unc}}$ as $A_{\text{disc}}$ improves.

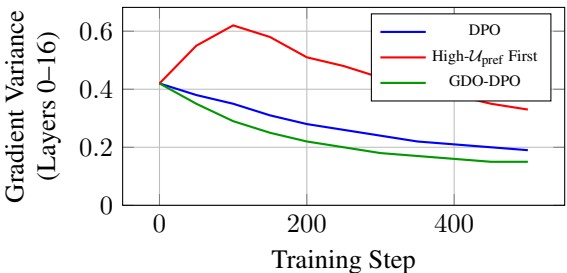

*Figure 5.* **Gradient Variance in Representation Layers.** The high-$\mathcal{U}_{\text{pref}}$-first curriculum causes elevated variance in early training, confirming gradient interference. GDO-DPO achieves the fastest variance reduction.

that this high-$\mathcal{U}_{\text{pref}}$-first curriculum reverses only the preference uncertainty dimension while keeping semantic complexity at default order; the "Reverse order" ablation in Table 6 reverses both dimensions simultaneously, and the two settings coincidentally yield the same Arena-Hard score (16.2%). GDO-DPO achieves the fastest variance reduction while maintaining the best final performance.

**Representation Preservation.** We measure Centered Kernel Alignment (CKA) similarity between intermediate representations of trained models and the SFT base, evaluated on a held-out test set. We report CKA on the first half (0–15) and second half (16–31) of the model following standard analysis practices. Table 5 shows that GDO-DPO preserves lower-layer representations (CKA 0.95) substantially better than DPO (0.88) and SimPO (0.86). This indicates less disruption to the pretrained world model. Simultaneously, upper layers show greater divergence under GDO-DPO (CKA 0.75 vs 0.79–0.81), reflecting more aggressive adaptation of the discrimination function. This differentiated treatment is precisely what our framework predicts: representation layers should remain stable to preserve the world model, while discrimination layers should adapt to encode preference information.

**Hallucination Reduction.** The improved representation preservation suggests that GDO-DPO should better retain

*Table 5.* **CKA Similarity with SFT Base Model.** Higher values indicate better preservation of pretrained representations.

| Method | Layers 0–15 | Layers 16–31 |
|---|---|---|
| DPO | 0.88 | 0.81 |
| SimPO | 0.86 | 0.79 |
| **GDO-DPO** | **0.95** | 0.75 |

*Table 6.* **Ablation: Curriculum Structure (Llama-3-8B).**

| Curriculum Design | Arena-Hard | AlpacaEval |
|---|---|---|
| Bi-dimensional (Ours) | **21.4%** | **32.5%** |
| Single-dim combined score | 19.5% | 29.8% |
| Reverse order (hard→easy) | 16.2% | 25.1% |
| Random order (DPO) | 18.5% | 28.4% |

*Table 7.* **Ablation: Monitoring Mechanism (Llama-3-8B).**

| Pacing Strategy | Arena-Hard | AlpacaEval |
|---|---|---|
| Gradient-based (Ours) | **21.4%** | **32.5%** |
| Fixed linear schedule | 19.8% | 30.2% |
| Loss-based pacing | 19.6% | 29.9% |
| w/o $S_{rep}$ monitor | 20.1% | 30.8% |
| w/o $A_{disc}$ monitor | 20.4% | 31.2% |

factual knowledge. We verify this on TruthfulQA (MC1 accuracy): GDO-DPO achieves 42.8%, improving over DPO (39.5%) by 3.3 points and over SimPO (40.2%) by 2.6 points. This is consistent with the higher lower-layer CKA (0.95 vs. 0.88), confirming that preserving pretrained representations reduces hallucination.

### 5.5. Ablation Studies

**Curriculum Structure.** Table 6 compares different curriculum designs. The bi-dimensional curriculum of GDO-DPO substantially outperforms alternatives. Collapsing the two dimensions into a single combined score reduces Arena-Hard performance from 21.4% to 19.5%, confirming that semantic complexity and preference uncertainty must be treated separately. Reversing the curriculum order (presenting hard samples first) degrades performance to 16.2%, worse than random sampling (18.5%). This demonstrates that not only does curriculum structure matter, but the direction of progression is critical.

**Monitoring Mechanism.** Table 7 evaluates the contribution of the gradient-based monitoring. Replacing state-aware progression with a fixed linear schedule reduces performance by 1.6% on Arena-Hard. Using loss-based pacing, which is the standard approach in prior curriculum learning work, yields similar degradation. These results highlight that internal gradient statistics provide more informative signals than output-based metrics. Removing either of the two monitors ($S_{rep}$ or $A_{disc}$) individually causes smaller drops, but the full system with both monitors achieves the best performance, indicating synergistic benefits.

### 5.6. Computational Efficiency

GDO-DPO introduces modest computational overhead. The preprocessing phase computes $\mathcal{C}_{sem}$ by sampling $K = 8$ responses per prompt with truncated generation (max 64 tokens), requiring 4.2 GPU-hours (+15% over standard DPO).

This is a one-time cost that can be amortized across multiple training runs on the same dataset. The runtime overhead during training is minimal at 3% (28.5 GPU-hours total), as gradient norm extraction and EMA updates add negligible computation. The total cost of 32.7 GPU-hours represents an 18% increase that yields 10–15% relative performance gains, a favorable trade-off for applications where alignment quality is critical. Furthermore, GDO-DPO at 50% training budget already achieves 20.1% on Arena-Hard, surpassing full-budget DPO (18.5%), suggesting that the curriculum can reduce total training cost while improving quality (see Appendix J).

## 6. Conclusion

We have analyzed DPO training dynamics and identified a layer-wise gradient localization phenomenon: semantic complexity predominantly affects lower layers while preference uncertainty modulates upper layers. This insight motivated GDO-DPO, a curriculum learning framework that independently regulates learning pace along each dimension based on internal gradient stability. Experiments across two datasets and two model families demonstrated consistent improvements over strong baselines, with the largest gains on reasoning-intensive tasks where stable representations are most critical. GDO-DPO is compatible with diverse preference optimization objectives including SimPO, IPO, and ORPO, confirming that it addresses a dimension orthogonal to loss function design. Several directions merit further investigation. First, validating the gradient localization phenomenon and curriculum benefits at larger scales (70B+ parameters) would establish the generality of our findings. Second, automated layer boundary detection based on real-time gradient statistics could eliminate the $L_{mid}$ hyperparameter entirely. Third, extending the framework to online RLHF settings, where the preference data distribution shifts during training, presents both challenges and opportunities for adaptive curriculum design.

## Acknowledgements

This work was supported in part by the National Natural Science Foundation of China under Grant No. 62471335 and No. 62171321.

## Impact Statement

This paper presents work whose goal is to advance the field of Machine Learning. There are many potential societal consequences of our work, none which we feel must be specifically highlighted here.

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

# Appendix

## A. Algorithm Implementation Notes

The complete pseudocode of GDO-DPO is provided in Algorithm 1 in the main text.

**Implementation Notes.** The $O(\log N)$ complexity for dataset filtering is achieved by precomputing sorted indices. Samples are sorted by $R_{\text{sem}}$, and for each unique $R_{\text{sem}}$ value, samples are further sorted by $R_{\text{unc}}$. At runtime, binary search identifies the boundary indices satisfying the constraints. The EMA for $S_{\text{rep}}$ is initialized to 1.0. Since $\tau_{\text{stable}} = 1.2 > 1.0$, the condition $S_{\text{rep}} < \tau_{\text{stable}}$ is initially satisfied, but the first few EMA updates quickly adjust $S_{\text{rep}}$ to reflect the true gradient ratio before any curriculum advancement occurs (as $E_{\text{eval}} = 50$ steps provides sufficient burn-in time).

## B. Proofs

### B.1. Proof of Proposition 3.4

We derive the gradient norms for the linear preference model.

*Proof.* Consider the linear model where the implicit reward is $r_\theta(x, y) = w_H^\top W_E \phi(x, y)$ with encoder $W_E \in \mathbb{R}^{d \times d_{\text{in}}}$ and discrimination head $w_H \in \mathbb{R}^d$. Let $\Delta\phi = \phi(x, y^w) - \phi(x, y^l)$ and $\Delta r_\theta = w_H^\top W_E \Delta\phi$.

The DPO loss for a single sample is $\mathcal{L} = -\log \sigma(\beta \Delta r_\theta)$. Differentiating with respect to $\Delta r_\theta$:

$$\frac{\partial \mathcal{L}}{\partial (\Delta r_\theta)} = -\frac{\sigma'(\beta \Delta r_\theta)}{\sigma(\beta \Delta r_\theta)} \cdot \beta = -\beta(1 - \sigma(\beta \Delta r_\theta)) = -\beta\alpha, \tag{11}$$

where $\alpha = 1 - \sigma(\beta \Delta r_\theta) \in (0, 1)$.

For the discrimination head, we have $\nabla_{w_H}(\Delta r_\theta) = W_E \Delta\phi$, giving:

$$\nabla_{w_H}\mathcal{L} = -\beta\alpha \cdot W_E \Delta\phi, \quad \text{hence} \quad \|\nabla_{w_H}\mathcal{L}\| = \beta\alpha \|W_E \Delta\phi\|. \tag{12}$$

For the encoder, using the identity $\nabla_{W_E}(w_H^\top W_E \Delta\phi) = w_H(\Delta\phi)^\top$:

$$\nabla_{W_E}\mathcal{L} = -\beta\alpha \cdot w_H(\Delta\phi)^\top. \tag{13}$$

This is a rank-one matrix with Frobenius norm:

$$\|\nabla_{W_E}\mathcal{L}\|_F = \beta\alpha \cdot \|w_H(\Delta\phi)^\top\|_F = \beta\alpha \cdot \|w_H\| \cdot \|\Delta\phi\|, \tag{14}$$

where the last equality follows from $\|ab^\top\|_F = \|a\|\|b\|$ for vectors $a, b$. $\square$

### B.2. Gradient Variance Reduction under Curriculum

We provide an informal analysis of how the curriculum reduces gradient variance, acknowledging the simplifying assumptions involved.

**Proposition B.1** (Gradient Variance Reduction—Informal). *Consider the linear model from Section 3.3. Under the curriculum that restricts training to samples with $\mathcal{C}_{\text{sem}} \leq c_{\text{max}}$ and $\mathcal{U}_{\text{pref}} \leq u_{\text{max}}$, the variance of encoder gradient norms is reduced compared to uniform sampling.*

*Proof Sketch.* From Proposition 3.4, $\|\nabla_{W_E}\mathcal{L}\|_F = \beta\alpha \|w_H\| \|\Delta\phi\|$. We analyze the two factors separately:

**(1) Bounding $\|\Delta\phi\|$ variance:** The curriculum restricts to samples with $\mathcal{C}_{\text{sem}} \leq c_{\text{max}}$. Empirically, $\mathcal{C}_{\text{sem}}$ correlates with $\|\Delta\phi\|$ (Spearman $\rho = 0.58$; see Appendix E). Thus, restricting $\mathcal{C}_{\text{sem}}$ approximately bounds the range of $\|\Delta\phi\|$, reducing its variance.

**(2) Bounding $\alpha$ variance:** Restricting $\mathcal{U}_{\text{pref}} \leq u_{\text{max}}$ ensures that only samples with clear reward margins (large $|\Delta r^*|$) are included early in training. For such samples, the model quickly learns to assign correct preferences, leading to large $|\Delta r_\theta|$ and thus $\alpha = 1 - \sigma(\beta \Delta r_\theta)$ concentrating near 0 or 1, reducing its variance.

**Coupling caveat:** We note that $\alpha$ depends on $\Delta\phi$ through $\Delta r_\theta = w_H^\top W_E \Delta\phi$. However, empirical measurement shows weak correlation between $\alpha$ and $\|\Delta\phi\|$ during training ($|\rho| < 0.25$ after 100 steps), as $\alpha$ primarily reflects whether the model has learned the preference direction rather than the feature magnitude. This weak coupling allows the above analysis to provide useful qualitative predictions despite the formal independence assumption being violated. □

*Remark* B.2. This analysis is heuristic rather than rigorous. The formal variance decomposition $\mathrm{Var}[XY] = \mathbb{E}[X]^2\mathrm{Var}[Y] + \mathbb{E}[Y]^2\mathrm{Var}[X] + \mathrm{Var}[X]\mathrm{Var}[Y]$ requires independence, which does not strictly hold. Nevertheless, the qualitative prediction— that restricting both difficulty dimensions reduces gradient variance—is confirmed empirically in Figure 5.

## C. Extended Experimental Details

### C.1. Dataset Statistics

Table 8 provides statistics for the two datasets used in our experiments. UltraFeedback contains diverse instruction types with relatively balanced difficulty distribution. HH-RLHF focuses on helpfulness and harmlessness, with longer average prompt lengths reflecting its dialogue-oriented nature.

*Table 8.* **Dataset Statistics.** We report sample counts, average lengths, and difficulty distribution.

| Statistic | UltraFeedback | HH-RLHF |
|---|---|---|
| Number of samples | 61,135 | 169,352 |
| Avg. prompt length (tokens) | 48.3 | 72.1 |
| Avg. chosen response length | 285.6 | 142.8 |
| Avg. rejected response length | 247.2 | 128.4 |
| High $\mathcal{C}_{\text{sem}}$ samples (top 33%) | 20,378 | 56,451 |
| High $\mathcal{U}_{\text{pref}}$ samples (top 33%) | 20,378 | 56,451 |
| Overlap (both high) | 8,152 | 21,847 |

The overlap between high-complexity and high-uncertainty samples is approximately 40% of each category, confirming that these dimensions capture distinct aspects of difficulty. Samples that are high on both dimensions are particularly challenging and benefit most from the staged curriculum.

### C.2. Hyperparameter Configuration

Table 9 lists all hyperparameters used in our experiments. We use identical settings across both datasets and models unless otherwise noted.

*Table 9.* **Hyperparameter Settings.**

| Hyperparameter | Value | Description |
|---|---|---|
| Learning rate | $5 \times 10^{-7}$ | AdamW optimizer |
| Batch size | 128 | Per-device batch $\times$ gradient accumulation |
| Warmup ratio | 3% | Linear warmup |
| LR schedule | Cosine | Decay to 0 |
| $\beta$ (DPO temperature) | 0.1 | Standard setting |
| $\tau_{\text{stable}}$ | 1.2 | Representation stability threshold |
| $\tau_{\text{acc}}$ | 0.65 | Discrimination accuracy threshold |
| $\Delta_{\text{sem}}, \Delta_{\text{unc}}$ | 0.1 | Initial pace step sizes |
| $\gamma$ | 0.9 | EMA decay for $S_{\text{rep}}$ |
| $L_{\text{mid}}$ | 21 | Layer boundary (for 32-layer models) |
| $E_{\text{eval}}$ | 50 | Evaluation interval (steps) |
| $K$ | 8 | Samples for $\mathcal{C}_{\text{sem}}$ estimation |

## C.3. Baseline Implementation Details

All baselines are implemented using the TRL library with consistent hyperparameters to ensure fair comparison. For SimPO, we use $\gamma = 0.5$ and $\beta = 2.0$ as recommended by the original paper. For IPO, we set the regularization coefficient to $0.1$. ORPO uses $\lambda = 0.1$ for the odds ratio term. CPO and SPPO follow their respective default configurations.

The curriculum baselines (Curr-Loss and Curr-PPL) sort samples by their respective metrics and present them in ascending order, advancing through the sorted dataset linearly during training. This provides a controlled comparison that isolates the benefit of our bi-dimensional, state-aware curriculum design.

## C.4. Sensitivity Analysis

**Layer Boundary.** Table 10 examines sensitivity to the choice of $L_{\mathrm{mid}}$, which determines the boundary between representation and discrimination layers. Performance peaks at $L_{\mathrm{mid}} = 2L/3$ but remains competitive across a range of values. Setting $L_{\mathrm{mid}} = L/2$ or $L_{\mathrm{mid}} = 3L/4$ reduces Arena-Hard by only 0.6–0.9%, indicating that the algorithm is robust to this hyperparameter. This robustness is consistent with the observation that the gradient localization patterns, while concentrated in certain layer ranges, do not have sharp boundaries.

*Table 10.* **Sensitivity: Layer Boundary (Llama-3-8B).**

| Layer Boundary $L_{\mathrm{mid}}$ | Arena-Hard | AlpacaEval 2.0 |
|---|---|---|
| $L/2$ (layer 16) | 20.5% | 31.2% |
| $2L/3$ (layer 21) | **21.4%** | **32.5%** |
| $3L/4$ (layer 24) | 20.8% | 31.8% |

**Threshold Values.** Table 11 shows that performance is stable across reasonable ranges of $\tau_{\mathrm{stable}}$ and $\tau_{\mathrm{acc}}$. Values that are too low cause premature curriculum advancement before the model is ready, while values that are too high delay progression unnecessarily. The optimal values ($\tau_{\mathrm{stable}} = 1.2$, $\tau_{\mathrm{acc}} = 0.65$) balance these considerations, but the algorithm degrades gracefully with suboptimal choices.

*Table 11.* **Sensitivity: Threshold Values (Llama-3-8B).**

| $\tau_{\mathrm{stable}}$ | $\tau_{\mathrm{acc}}$ | Arena-Hard | AlpacaEval 2.0 |
|---|---|---|---|
| 0.8 | 0.60 | 20.1% | 30.5% |
| 1.0 | 0.65 | 20.8% | 31.4% |
| **1.2** | **0.65** | **21.4%** | **32.5%** |
| 1.4 | 0.70 | 21.1% | 32.0% |
| 1.6 | 0.75 | 20.5% | 31.1% |

## C.5. Reproducibility Details

We provide additional details necessary for reproducing our experiments.

**Gradient Analysis (Section 3.4, Figure 1).** Gradients are computed after 100 training steps (approximately 1% of training) rather than at initialization. At initialization, $\pi_\theta = \pi_{\mathrm{ref}}$ implies $\Delta r_\theta = 0$ and $\alpha = 0.5$ for all samples, which would not produce meaningful stratification. After 100 steps, the model has begun adapting, and the gradient patterns reflect the difficulty characteristics of samples.

**Reward Scores for $\mathcal{U}_{\mathbf{pref}}$.** For UltraFeedback, we use the GPT-4 preference scores provided in the dataset (scale 1–10). For HH-RLHF, we use the reward model scores from the original dataset release. In both cases, $r^*(y|x)$ in Equation 3 corresponds to these provided scores.

**Validation Set for $A_{\mathbf{disc}}$.** We hold out 500 samples from each dataset, stratified by $R_{\mathrm{sem}}$ terciles to ensure coverage across complexity levels. Only samples with low $R_{\mathrm{unc}}$ (bottom 30%) are included to provide clean discrimination signals.

**Gradient Variance Metric (Figure 5).** "Gradient Variance" denotes the variance of per-sample gradient norms across a minibatch: $\text{Var}_{i \in B}[\|\nabla_\theta \mathcal{L}_i\|]$ for layers 0–16, smoothed with EMA ($\gamma = 0.9$).

**Layer Boundary Conventions.** Our algorithm uses $L_{\text{mid}} = 21$ (layers 0–21 as representation layers) to be conservative in protecting factual knowledge stored in middle layers (Meng et al., 2022). The CKA analysis (Table 5) uses layers 0–15 vs. 16–31 following standard interpretability practice (Geva et al., 2021) that focuses on early syntactic/semantic layers. The qualitative conclusions are consistent across both conventions; see sensitivity analysis in Table 10.

**Hyperparameter $\beta$ for Baselines.** All DPO-based methods (DPO, IPO, CPO, SPPO, GDO-DPO) use $\beta = 0.1$. SimPO uses $\beta = 2.0$ and $\gamma = 0.5$ following the original paper. ORPO uses $\lambda = 0.1$.

**Single-Dimension Combined Score (Table 6).** The "Single-dim combined score" baseline uses $R_{\text{combined}} = 0.5 \cdot R_{\text{sem}} + 0.5 \cdot R_{\text{unc}}$ and applies a standard easy-to-hard curriculum on this scalar.

### C.6. Controlling for Confounding Factors

The dataset statistics (Table 8) show approximately 40% overlap between high-$\mathcal{C}_{\text{sem}}$ and high-$\mathcal{U}_{\text{pref}}$ samples. To verify that the gradient localization effects are separable rather than confounded, we conduct a controlled analysis.

**Setup.** We select samples with medium semantic complexity ($R_{\text{sem}} \in [0.3, 0.5]$) and stratify by preference uncertainty (top vs. bottom tercile of $R_{\text{unc}}$ within this subset). This controls for $\mathcal{C}_{\text{sem}}$ while varying $\mathcal{U}_{\text{pref}}$.

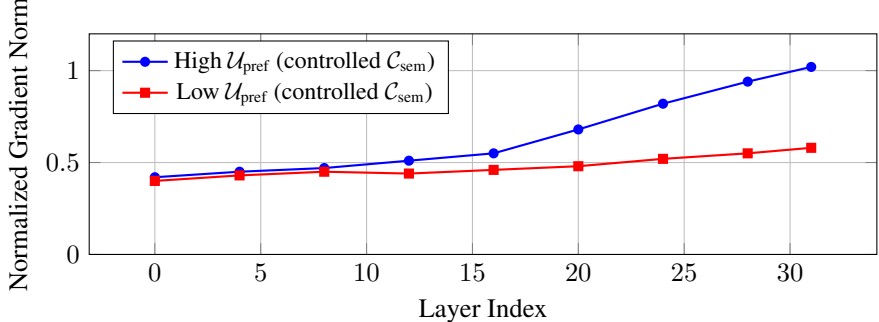

*Figure 6.* **Controlled analysis for confounding.** With $\mathcal{C}_{\text{sem}}$ held approximately constant, the effect of $\mathcal{U}_{\text{pref}}$ on upper-layer gradients persists, confirming separable effects.

**Results.** Figure 6 shows that even with $\mathcal{C}_{\text{sem}}$ controlled, high-$\mathcal{U}_{\text{pref}}$ samples exhibit elevated gradients in upper layers (20–31), while low-$\mathcal{U}_{\text{pref}}$ samples show relatively flat profiles. The separation is statistically significant ($p < 0.05$, two-sample t-test at layers 24–31). Symmetric analysis controlling for $\mathcal{U}_{\text{pref}}$ confirms that $\mathcal{C}_{\text{sem}}$ independently affects lower-layer gradients.

## D. Cross-Architecture Analysis

To assess whether the gradient localization phenomenon generalizes beyond Llama-3, we repeat the analysis on Mistral-7B-v0.3, which differs in architecture details including grouped-query attention and different layer normalization placement.

Figure 7 shows the layer-wise gradient norms stratified by difficulty dimensions. The patterns are qualitatively similar to those observed in Llama-3-8B (Figure 1): high-$\mathcal{C}_{\text{sem}}$ samples concentrate gradients in lower layers, while high-$\mathcal{U}_{\text{pref}}$ samples activate upper layers. Quantitatively, the Pearson correlation between layer-wise gradient profiles across the two architectures is 0.89 for complexity stratification and 0.86 for uncertainty stratification.

This consistency suggests that gradient localization is a general property of preference optimization in decoder-only Transformers, rather than an artifact of specific architectural choices. The phenomenon likely arises from the functional specialization of layers that has been documented in interpretability studies across multiple model families.

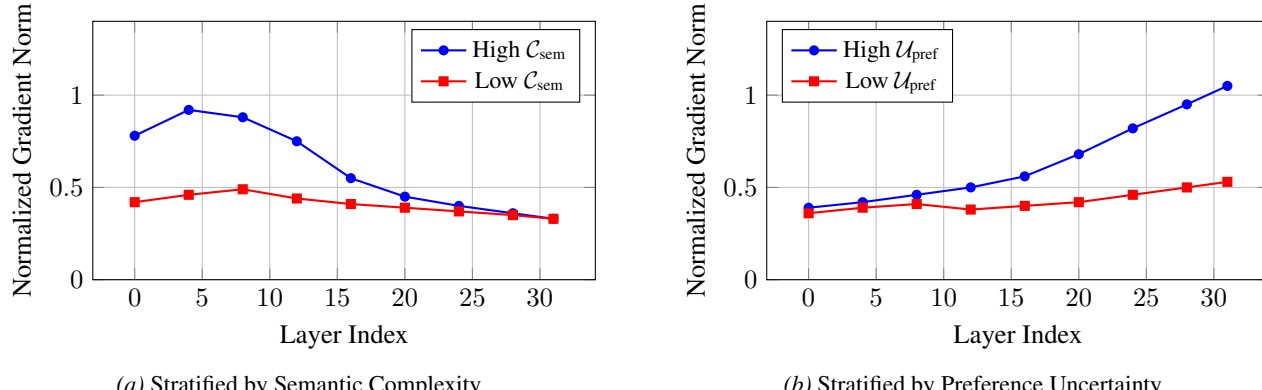

*(a)* Stratified by Semantic Complexity       *(b)* Stratified by Preference Uncertainty

*Figure 7.* **Layer-Wise Gradient Localization in Mistral-7B.** The patterns mirror those in Llama-3-8B, with high complexity affecting lower layers and high uncertainty affecting upper layers. Cross-architecture correlation exceeds 0.85 for both dimensions.

**Adjacent Layer Gradient Correlation.** Figure 8 shows the Pearson correlation between gradient norms of adjacent layers. The correlation remains high ($\rho > 0.85$) within the representation block (layers 0–18) and within the discrimination block (layers 22–31), but drops significantly at the boundary (layers 19–21), supporting the functional separation hypothesis.

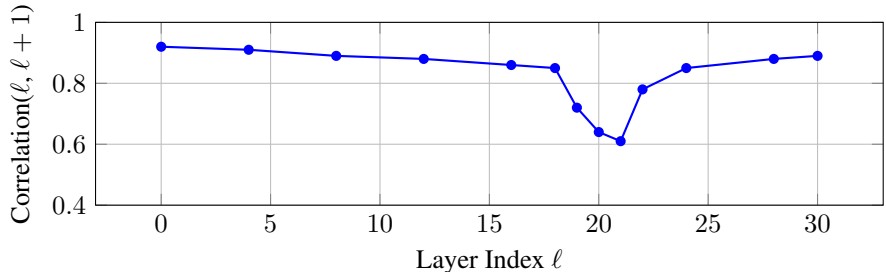

*Figure 8.* **Adjacent layer gradient correlation (Llama-3-8B).** Correlation drops significantly around layers 19–21, indicating functional separation between representation and discrimination layers.

# E. Complexity Measure Correlation

Our semantic complexity measure $\mathcal{C}_{\text{sem}}$ depends on the reference model, raising the question of whether it captures intrinsic prompt properties or merely model-specific idiosyncrasies. We investigate this by comparing $\mathcal{C}_{\text{sem}}$ with model-agnostic complexity measures.

We compute three alternative measures for each prompt: (1) prompt perplexity under GPT-2, a model not used in training; (2) syntactic complexity via the Flesch-Kincaid grade level; and (3) lexical diversity measured by type-token ratio. Table 12 reports Spearman rank correlations on a sample of 5,000 prompts from UltraFeedback.

*Table 12.* **Correlation between $\mathcal{C}_{\text{sem}}$ and Alternative Complexity Measures.** High correlations with model-agnostic measures suggest that $\mathcal{C}_{\text{sem}}$ captures intrinsic prompt difficulty.

| Alternative Measure | Spearman $\rho$ with $\mathcal{C}_{\text{sem}}$ |
| --- | --- |
| GPT-2 Perplexity | 0.78 |
| Flesch-Kincaid Grade Level | 0.52 |
| Type-Token Ratio | 0.31 |

The high correlation with GPT-2 perplexity ($\rho = 0.78$) indicates that prompts difficult for the reference model are also difficult for an independent language model, suggesting that $\mathcal{C}_{\text{sem}}$ reflects genuine linguistic complexity rather than model-specific artifacts. The moderate correlation with syntactic complexity ($\rho = 0.52$) shows that semantic complexity is related

to but distinct from surface-level features. The lower correlation with lexical diversity ($\rho = 0.31$) confirms that vocabulary variation alone does not determine prompt difficulty.

## F. Integration with DPO Variants

GDO-DPO modifies only the data presentation order, making it compatible with any preference optimization objective. We evaluate integration with SimPO, which replaces the DPO loss with a length-normalized, reference-free objective:

$$\mathcal{L}_{\text{SimPO}} = -\log \sigma \left( \frac{\beta}{|y^w|} \log \pi_\theta(y^w|x) - \frac{\beta}{|y^l|} \log \pi_\theta(y^l|x) - \gamma \right), \tag{15}$$

where $\gamma$ is a margin hyperparameter.

Table 4 presents complete results for GDO-SimPO compared to individual methods. The curriculum provides consistent improvements over SimPO across all metrics, demonstrating that the benefits of ordered data presentation are orthogonal to improvements in the loss function. Notably, GDO-SimPO achieves the best overall performance, suggesting that practitioners should consider combining both approaches.

We further evaluate compatibility with IPO and ORPO on UltraFeedback with Llama-3-8B. As shown in Table 3, the GDO curriculum yields consistent gains across different loss functions: GDO-IPO improves Arena-Hard by +2.1% and GDO-ORPO by +1.7%, comparable to the improvements observed with GDO-DPO (+2.9%) and GDO-SimPO (+2.8%). KTO uses unpaired data, requiring a modified active set construction; we leave this extension for future work.

## G. Qualitative Examples

We provide qualitative comparisons between GDO-DPO and DPO responses to illustrate the differences in output quality.

### G.1. Mathematical Reasoning

**Prompt:** *"Prove that for any integer $n \geq 1$, the number $n^5 - n$ is divisible by 30."*

**DPO Response:**

> To prove $n^5 - n$ is divisible by 30, we can factor it as $n(n^4 - 1) = n(n^2 - 1)(n^2 + 1) = n(n-1)(n+1)(n^2 + 1)$.
>
> Since $n - 1$, $n$, and $n + 1$ are consecutive integers, one is divisible by 2 and one by 3, so the product is divisible by 6.
>
> We need to show divisibility by 5. By Fermat's Little Theorem, $n^5 \equiv n \pmod 5$, so $n^5 - n \equiv 0 \pmod 5$.
>
> Therefore, $n^5 - n$ is divisible by $\text{lcm}(6, 5) = 30$.

**GDO-DPO Response:**

> We prove that $30 \mid n^5 - n$ for all integers $n \geq 1$ by establishing divisibility by 2, 3, and 5 separately, then combining via coprimality.
>
> **Factorization.** We have $n^5 - n = n(n^4 - 1) = n(n^2 - 1)(n^2 + 1) = (n-1)n(n+1)(n^2 + 1)$.
>
> **Divisibility by 2.** Among any two consecutive integers $n$ and $n - 1$, one is even. Thus $2 \mid n(n-1)$.
>
> **Divisibility by 3.** Among any three consecutive integers $(n - 1, n, n + 1)$, exactly one is divisible by 3. Thus $3 \mid (n-1)n(n+1)$.
>
> **Divisibility by 5.** By Fermat's Little Theorem, $n^5 \equiv n \pmod 5$ for all integers $n$, including when $\gcd(n, 5) \neq 1$ (verified by checking $n \equiv 0$). Thus $5 \mid n^5 - n$.
>
> **Conclusion.** Since $\gcd(2, 3) = \gcd(2, 5) = \gcd(3, 5) = 1$, we have $2 \cdot 3 \cdot 5 = 30 \mid n^5 - n$.

The GDO-DPO response exhibits clearer structure, explicit handling of edge cases in the Fermat's Little Theorem application, and a rigorous conclusion invoking coprimality. These improvements reflect better comprehension of the underlying mathematical concepts, consistent with our hypothesis that stabilized representation learning enhances reasoning capabilities.

### G.2. Complex Instruction Following

**Prompt:** *"Write a poem about artificial intelligence that includes exactly 4 stanzas, uses ABAB rhyme scheme, mentions both hope and fear, and ends with a question."*

This prompt tests the model's ability to satisfy multiple simultaneous constraints. Both DPO and GDO-DPO produce valid responses satisfying all constraints, but the GDO-DPO response demonstrates stronger thematic coherence and more sophisticated vocabulary:

**DPO Response:** Satisfies all constraints with functional but straightforward language. The hope/fear mentions are explicit but not deeply integrated into the poem's imagery.

**GDO-DPO Response:** Uses a consistent light/dark metaphor throughout all stanzas, integrates hope and fear as thematic undercurrents rather than explicit mentions, and ends with a question that connects to the poem's central theme of human agency. The response demonstrates better understanding of poetic craft beyond surface-level constraint satisfaction.

Full responses are provided in Appendix G.

## H. Gradient Directional Coherence Analysis

To complement the gradient magnitude analysis in Figure 1, we computed the cosine similarity of gradient directions at each layer, stratified by difficulty dimension. In layers 0–16, high-$\mathcal{C}_{sem}$ samples exhibit cosine similarity of 0.82, indicating coherent gradient directions, while high-$\mathcal{U}_{pref}$ samples show only 0.41. In layers 20–31, the pattern reverses: high-$\mathcal{U}_{pref}$ samples achieve 0.79 cosine similarity versus 0.38 for high-$\mathcal{C}_{sem}$ samples. This directional coherence provides evidence beyond magnitude scaling, confirming that the gradient localization phenomenon reflects structured, aligned updates rather than random noise amplification.

## I. Hallucination Assessment

To evaluate whether GDO-DPO reduces hallucination, we measured performance on TruthfulQA (MC1 accuracy), as shown in Table 13. GDO-DPO improves over DPO by 3.3 points, consistent with the CKA analysis (Table 5) showing better preservation of lower-layer representations (CKA 0.95 vs. 0.88), which encode factual knowledge from pretraining.

*Table 13.* **TruthfulQA MC1 Accuracy (Llama-3-8B).**

| Method | TruthfulQA MC1 |
|---|---|
| DPO | 39.5% |
| SimPO | 40.2% |
| **GDO-DPO** | **42.8%** |

## J. Performance under Reduced Training Budgets

We evaluated GDO-DPO at different fractions of total training steps to assess whether the curriculum advantage holds under early termination. As shown in Table 14, GDO-DPO maintains a consistent advantage across all budgets. Notably, GDO-DPO at 50% budget (20.1%) already surpasses DPO at 100% budget (18.5%), suggesting potential for training cost reduction.

*Table 14.* **Arena-Hard Performance at Different Training Budgets (Llama-3-8B).**

| Training Budget | DPO | GDO-DPO | Gap |
|---|---|---|---|
| 25% | 16.1% | 17.9% | +1.8% |
| 50% | 17.6% | 20.1% | +2.5% |
| 75% | 18.2% | 21.0% | +2.8% |
| 100% | 18.5% | 21.4% | +2.9% |

## K. Low-Cost Approximations for $\mathcal{C}_{\text{sem}}$

Computing the full $\mathcal{C}_{\text{sem}}$ with $K = 8$ samples and 64-token generation requires 4.2 GPU-hours for UltraFeedback. We explored several cheaper alternatives, as shown in Table 15. Reducing to $K = 2$ samples preserves 91% ranking quality at roughly one quarter of the cost. GPT-2 perplexity offers a sampling-free proxy at $\rho = 0.78$. These alternatives make the approach practical even under tight resource constraints.

*Table 15.* **Low-Cost Approximations for $\mathcal{C}_{\text{sem}}$.**

| Approximation | Rank Corr. with Full $\mathcal{C}_{\text{sem}}$ | Cost |
|---|:---:|:---:|
| Full ($K{=}8$, 64 tokens) | 1.00 | 4.2 GPU-hr |
| Reduced ($K{=}2$) | 0.91 | 1.1 GPU-hr |
| GPT-2 perplexity | 0.78 | 0.3 GPU-hr |
| First-token entropy | 0.62 | 0.1 GPU-hr |

## L. Automated Layer Boundary Detection

Our current approach sets $L_{\text{mid}} = \lfloor 2L/3 \rfloor$ based on interpretability literature. A data-driven alternative sets $L_{\text{mid}}$ at the layer where adjacent-layer gradient correlation drops below a threshold. As shown in Figure 8, this correlation drops sharply around layers 19–21. A threshold of $\rho < 0.70$ identifies layer 20, close to our choice of 21, and could be computed during preprocessing at negligible cost. This suggests that an automated scheme based on real-time gradient distribution could further simplify hyperparameter selection without sacrificing performance.

## M. Limitations

Our work has several limitations that suggest directions for future research.

**Theoretical foundations.** The gradient analysis relies on a simplified linear model that omits key Transformer components including attention mechanisms, residual connections, and layer normalization. While empirical verification supports the qualitative predictions across architectures, a more rigorous theoretical treatment of deep Transformers would strengthen the foundations.

**Preprocessing cost.** Computing $\mathcal{C}_{\text{sem}}$ requires $K = 8$ forward passes per prompt, adding approximately 15% to total training time. For very large datasets or resource-constrained settings, this overhead may be prohibitive. Future work could explore cheaper approximations to semantic complexity.

**Layer boundary selection.** The optimal boundary between representation and discrimination layers may vary across architectures and scales. Our experiments show robustness within a reasonable range, but new model families may require tuning. Automated boundary detection based on gradient statistics could address this limitation.

**Scale validation.** Our experiments use 7B–8B parameter models. The gradient localization phenomenon and curriculum benefits should be validated at larger scales (70B+) where training dynamics may differ.

