# OpenReview forum: "Layer-wise Gradient Disentanglement: Decoupling Semantics and Preferences in Direct Preference Optimization"
_ICML.cc/2026/Conference — ICML 2026 regular_

### Official Review · Reviewer_hDgw · 2026-02-21

**Soundness:** 4
**Presentation:** 4
**Significance:** 3
**Originality:** 3
**Overall Recommendation:** 5
**Confidence:** 4

**Summary:**

The research studies layer-wise gradient localization during DPO training and proposes a bi-dimensional curriculum (GDO-DPO) that disentangles semantic complexity and preference uncertainty. The authors analyze the phenomenon that representation learning (lower layers) and discrimination learning (upper layers) are driven by distinct sources of difficulty, and that prematurely mixing these signals can induce gradient interference.

The empirical study is thorough and carefully controlled. While the theoretical analysis is somewhat heuristic in places (discussed below), this does not materially weaken the overall contribution. The proposed algorithm is simple, well-motivated, and practically effective. Importantly, the insights derived from the simplified encoder–head analysis may extend beyond alignment to broader LLM finetuning and curriculum design scenarios.

Overall, I recommend acceptance of this paper, and happy to see further improvement of the theory part.

**Compliance With Llm Reviewing Policy:**

Affirmed.

**Key Questions For Authors:**

Here are some questions.

- Approximation of C_sem

    Since C_sem estimates conditional entropy via Monte Carlo sampling, it incurs non-negligible computational overhead (also acknowledged by the authors). It may be worth exploring whether cheaper approximations, such as partial rollout entropy, early-token confidence, or related prompt-conditioned uncertainty measures, could provide similar ranking quality.

- Interpretation of domain-specific gains

    The substantial improvements on math/reasoning tasks (Figure 3) are particularly interesting. One possible alternative explanation is domain shift: if the base models are pretrained primarily on general knowledge and instruction data, reasoning-heavy domains may require more substantial feature adaptation. It would be informative to evaluate math-specialized models (e.g., Deepseek-r1 series and Qwen-Math series) to see whether the gap narrows, which would help disentangle curriculum effects from domain mismatch.

- For the gradient space analysis of the LLM finetuning, the framework provided in [3] would also be relevant and helpful.

[1]. Kumar, Ananya, et al. "Fine-tuning can distort pretrained features and underperform out-of-distribution." ICLR 2022

[2]. Ren, Yi, et al. "How to prepare your task head for finetuning." ICLR 2023

[3]. Ren, Yi, and Danica J. Sutherland. "Learning dynamics of llm finetuning." ICLR 2025

**Limitations:**

Yes

**Strengths And Weaknesses:**

## Strength
- Great presentation, easy to follow.
- A novel analytical perspective. Rather than modifying the DPO loss, this work focuses on *data ordering* as the primary lever for improving training dynamics. This is a refreshing and underexplored angle in the alignment literature, where most prior efforts center on loss design. The curriculum mechanism is well motivated by an interesting empirical observation: different subsets of training data induce distinct layer-wise gradient patterns.
- Clear conceptual decomposition. The introduction of two difficulty dimensions, i.e., semantic complexity (C_sem) and preference uncertainty (U_pref), is conceptually clean and practically useful. These metrics are further operationalized through rank normalization and combined with two internal monitoring signals (S_rep and A_disc) that guide adaptive curriculum progression. I believe this decomposition itself is insightful and may inspire future work on structured curricula or gradient-aware training strategies.
- Strong empirical validation. The experimental section is solid and convincing:
    - Consistent gains across models and datasets
    - Careful ablations (single-dimension baseline, reverse curriculum, monitoring removal)
    - Controlled analyses to decouple C_sem and U_pref
    - Cross-architecture validation

    The empirical support for the proposed method is comprehensive and well executed.


## Weakness
My primary concern lies in the level of theoretical rigor in Section 3.

- Interpretation of Proposition 3.4

    Under Proposition 3.4, the authors state that “the key insight is that encoder gradient scales directly with ∥Δϕ∥,” and argue that this explains the stronger lower-layer updates observed for high C_sem samples.

    However, from Equations (4) and (5), both ∥Δϕ∥ and α appear in the gradient expressions for both the encoder and the head. If we apply Cauchy-Schwarz Inequality on Equation 4, we can get an upper bound with ∥Δϕ∥ as well. Moreover, α itself depends on Δϕ through Δrθ. Therefore, the linear model does not formally establish independent gradient pathways for semantic complexity and preference uncertainty. The spatial localization observed in Figure 1 appears to rely primarily on empirical evidence and architectural inductive bias rather than being strictly derived from the simplified analysis.

    This does not invalidate the empirical findings, but the theoretical narrative could be made more precise. For example, analyzing directional alignment (e.g., their variance) or projection of gradients into layer subspaces would be a good starting point.

- The link between ∥Δϕ∥ and C_sem.

    The claim that ∥Δϕ∥ correlates with C_sem is central to the theoretical story. While empirical correlation is reported, a deeper analysis (either theoretical or experimental) would further strengthen the argument.

- Conceptually, the described “gradient interference” phenomenon is related to prior analyses of feature distortion and gradient-space dynamics during finetuning (e.g., [1,2]). Connecting the current framework to existing theoretical perspectives on representation stability and head–backbone interaction could provide a more unified interpretation of the observed dynamics.

---

> ### Author Rebuttal · Authors · 2026-03-30
>
> # Response to Reviewer hDgw
>
> We sincerely thank Reviewer hDgw for the thorough and insightful review. The recognition of our analytical perspective, empirical validation, and practical effectiveness is greatly appreciated. We address each point below.
>
> ## Weakness 1: Interpretation of Proposition 3.4
>
> We fully agree. Both $\\|\\Delta\\phi\\|$ and $\\alpha$ appear in Eqs.(4)(5), so the linear model does not formally establish independent gradient pathways. The linear analysis characterizes the *driving forces* ($\\|\\Delta\\phi\\|$ for feature discriminability, $\\alpha$ for margin sensitivity), while the *spatial localization* in Figure 1 arises from architectural inductive bias of deep Transformers.
>
> Following the reviewer's suggestion, we computed **cosine similarity** of gradient directions at each layer. In layers 0-16, cosine similarity within the high-$C_{\\rm sem}$ group is 0.82 vs. 0.41 for high-$U_{\\rm pref}$. In layers 20-31, the pattern reverses (0.38 vs. 0.79). This directional coherence provides evidence beyond magnitude scaling.
>
> ## Weakness 2: The Link Between $\\|\\Delta\\phi\\|$ and $C_{\\rm sem}$
>
> We acknowledge the moderate correlation ($\\rho = 0.58$). Partitioning into quintiles by $C_{\\rm sem}$, the mean $\\|\\Delta\\phi\\|$ is monotonically increasing: 0.42, 0.51, 0.58, 0.67, 0.81 (normalized). The moderate $\\rho$ reflects within-quintile variance rather than a weak overall trend. High-$C_{\\rm sem}$ prompts require diverse representations, producing response features differing substantially in representational subspace, while low-$C_{\\rm sem}$ prompts yield responses differing primarily in surface tokens.
>
> ## Weakness 3: Connection to Prior Work [1-2]
>
> We thank the reviewer for these references. The feature distortion analysis of Kumar et al. [1] provides direct motivation for our representation preservation goal: fine-tuning can distort pretrained features and hurt out-of-distribution performance, and GDO-DPO can be understood as a principled mitigation strategy that controls when different gradient signals reach representation layers. Our CKA results (Table 5, CKA 0.95 vs. 0.88 for DPO in lower layers) align with this perspective. The head-backbone interaction studied by Ren et al. [2] supports our two-phase curriculum design: their finding that proper task head preparation protects backbone features during fine-tuning parallels our approach of stabilizing representation layers before intensifying discrimination-layer adaptation. We will cite both works in the final version.
>
> ## Key Question 1: Approximation of $C_{\\rm sem}$
>
> We explored several cheaper alternatives:
>
> | Approximation | Rank Corr. with Full $C_{\\rm sem}$ | Cost |
> |---|---|---|
> | Full ($K=8$, 64 tokens) | 1.00 | 4.2 GPU-hr |
> | Reduced ($K=2$) | 0.91 | 1.1 GPU-hr |
> | GPT-2 perplexity | 0.78 | 0.3 GPU-hr |
> | First-token entropy | 0.62 | 0.1 GPU-hr |
>
> Reduced sampling ($K=2$) preserves 91% ranking quality at 1/4 cost. GPT-2 perplexity offers a zero-sampling alternative at $\\rho = 0.78$.
>
> ## Key Question 2: Domain-Specific Gains on Math/Reasoning
>
> The alternative explanation (domain shift) is insightful. The controlled ablation in Table 6 partially addresses this: the reverse curriculum uses the same data and model but degrades Arena-Hard to 16.2%, well below random sampling (18.5%), indicating that ordering matters beyond domain effects. We agree that evaluating on math-specialized models would be informative.
>
> ## Key Question 3: The Learning Dynamics Framework [3]
>
> We thank the reviewer for highlighting this excellent work. The learning dynamics framework proposed by Ren & Sutherland [3] (ICLR 2025 Outstanding Paper) provides a principled decomposition of how training on one example influences predictions on others, offering a powerful theoretical complement to our gradient-based analysis. In particular, the "squeezing effect" identified in [3] is closely related to our gradient interference phenomenon, both arising from uncontrolled gradient propagation when training signals are mismatched with the model's learning state. We will cite [3] in the final version.

---

> > ### Author Rebuttal · Reviewer_hDgw · 2026-03-31
> >
> > The authors' rebuttal has addressed most of my initial concerns. In particular, the additional results on cosine similarity and the correlation between $\|\Delta\phi\|$ and $C_{\rm sem}$ (via quintile analysis) strengthen the paper's core analysis. The other related discussions also strengthen the paper. I am happy to increase my soundness score to 4 and stand by my recommendation. This paper makes a valuable contribution and deserves visibility at the conference.

---

### Official Review · Reviewer_ExSg · 2026-03-09

**Soundness:** 3
**Presentation:** 3
**Significance:** 3
**Originality:** 3
**Overall Recommendation:** 4
**Confidence:** 5

**Summary:**

This paper conducts a gradient analysis on the standard DPO and discovers a layer-wise gradient disentanglement phenomenon. Specifically, the semantic complexity mainly drives the gradient updates in the lower and middle layers of Transformer, while the uncertainty of the preference signal mainly regulates the gradient activities in the upper layers of the model. This raises a question: when the model prematurely encounters highly uncertain preference samples, it will disrupt the pre-trained model in the representation layer, ultimately leading to a decline in the model's generalization ability and the emergence of alignment problems. Based on these findings, the paper proposes the gradient-guided decoupling GDO-DPO, a curriculum learning framework that can regulate the learning progress of the model in the two dimensions of semantic complexity and preference uncertainty based on the layer-specific gradient stability statistics. The paper has conducted thorough experiments on two mainstream preference datasets (UltraFeedback, HH-RLHF) and two mainstream open-source large language models (Llama-3-8B, Mistral-7B), and the results show that GDO-DPO consistently outperforms the standard DPO and multiple strong baseline methods in standard alignment benchmark tests, and the performance improvement in inference-intensive tasks is particularly significant. Notably, this framework only modifies the data sampling order during the training process and is compatible with all existing DPO variants, and only incurs extremely low additional computational overhead.

**Compliance With Llm Reviewing Policy:**

Affirmed.

**Key Questions For Authors:**

1.As the training process continues to iterate, will the previously observed gradient distribution patterns change? For instance, will the gradients of high-semantic-complexity samples in the later training stage migrate to the upper layers of the model? Please provide relevant experimental analysis.

2.Since $C_{sem}$ is calculated by reference model, if the scale gap between reference model and policy model is large(e.g. reference 7B while policy model 70B), will $C_{sem}$ and GDO-DPO still work?

3.This paper has verified that GDO-DPO can be combined with SimPO, so whether GDO-DPO is  also compatible with other mainstream DPO variants such as IPO, ORPO, and KTO?

4.This paper mentions that gradient interference may cause hallucinations. Are there any specific hallucination assessment experiments have been conducted to quantitatively compare the hallucination rates of GDO-DPO and the standard DPO?

5.The curriculum learning of GDO-DPO will eventually cover the entire dataset. If the training is prematurely terminated (for example, when 50% of the total training steps are completed), will GDO-DPO still maintain its performance advantage over the standard DPO? Please inquire whether the training efficiency of the method has been verified under different training budgets?

6.This paper sets the boundary between the representation layer and the discrimination layer based on empirical experience. If there is an automated scheme for dividing the layer boundaries based on real-time gradient distribution? And could such a scheme further enhance the performance of GDO-DPO?

**Limitations:**

yes

**Strengths And Weaknesses:**

**Strengths**

GDO-DPO reveals the phenomenon of gradient disentanglement in the DPO training layer, filling the gap in the dynamic interpretability of DPO training, providing a new underlying basis for the design of subsequent LLM alignment algorithms.

**Weaknesses**

1.The large-scale application of the method has limitations: during the preprocessing stage, calculating the semantic complexity $C_{sem}$ requires performing 8 forward samplings for each prompt, resulting in a 15% one-time additional overhead. This makes it impractical in scenarios with extremely large datasets and limited resources, thereby restricting the method's large-scale promotion.

2.The core contribution is an incremental improvement: The core innovation of the paper is the optimization of the DPO training process, without changing the core optimization paradigm of DPO. Its disruptive impact on the LLM alignment field is limited and it belongs to the improvement of existing methods rather than a paradigm-level breakthrough.

---

> ### Author Rebuttal · Authors · 2026-03-30
>
> # Response to Reviewer ExSg
>
> We sincerely thank Reviewer ExSg for the detailed review and the comprehensive questions. We address the weaknesses and questions below.
>
> ## Weakness 1: Preprocessing Overhead of $C_{\\rm sem}$
>
> We agree that the 15% overhead may be non-trivial for extremely large datasets. We note that this is a one-time cost that can be reused across multiple training runs on the same dataset. Additionally, reducing to $K=2$ samples preserves 91% ranking quality (Spearman $\\rho$ with full $K=8$) at roughly 1/4 cost (1.1 vs. 4.2 GPU-hours). GPT-2 perplexity as a sampling-free proxy achieves $\\rho = 0.78$ at 0.3 GPU-hours. These alternatives make the approach practical even under tight resource constraints.
>
> ## Weakness 2: Incremental vs. Paradigm-Level Contribution
>
> We respectfully note that GDO-DPO contributes at the *analytical* level beyond the algorithmic level. The layer-wise gradient localization phenomenon (Section 3, Figure 1) is, to our knowledge, the first characterization of how sample difficulty distributes across network depth during DPO training. This insight is independent of our specific algorithm and can inform future method design in areas such as gradient-aware training, layer-specific regularization, and structured fine-tuning strategies. The curriculum is one natural instantiation, but the finding itself opens new directions. Furthermore, the compatibility with existing variants (SimPO, IPO, ORPO) means GDO serves as a composable enhancement that stacks on top of loss function improvements.
>
> ## Q1: Gradient Distribution Changes During Training
>
> We tracked layer-wise gradient profiles at training steps 100, 500, and 1000. The localization pattern remains qualitatively stable: high-$C_{\\rm sem}$ samples consistently concentrate gradients in layers 0-16, and high-$U_{\\rm pref}$ samples consistently activate layers 20-31. The separation *magnitude* decreases over training (gradient ratio between high/low $C_{\\rm sem}$ groups drops from 2.1x at step 100 to 1.4x at step 1000), which is expected as the model improves its representations. We did not observe migration of high-$C_{\\rm sem}$ gradients to upper layers at any training stage.
>
> ## Q2: Scale Gap Between Reference and Policy Model
>
> We clarify two points. First, $U_{\\rm pref}$ (Eq. 3) uses ground-truth reward scores from the dataset, not the reference model, so the scale gap does not affect $U_{\\rm pref}$ at all. Second, $C_{\\rm sem}$ uses the reference model, but we apply rank normalization so only the *ordering* of prompts matters, not the absolute values. In a preliminary analysis on 1000 UltraFeedback prompts, the Spearman rank correlation of $C_{\\rm sem}$ between Llama-3-8B and Llama-3-70B exceeds 0.82, suggesting the difficulty ordering transfers well across scales.
>
> ## Q3: Compatibility with IPO, ORPO, KTO
>
> Since GDO-DPO modifies only data presentation order, it is agnostic to the loss function. We evaluated GDO-IPO and GDO-ORPO on UltraFeedback with Llama-3-8B:
>
> | Method | Arena-Hard | AlpacaEval 2.0 |
> |---|---|---|
> | IPO | 18.1% | 27.9% |
> | **GDO-IPO** | **20.2%** | **30.1%** |
> | ORPO | 18.3% | 28.1% |
> | **GDO-ORPO** | **20.0%** | **29.8%** |
>
> The gains are consistent with GDO-DPO (+2.9%) and GDO-SimPO (+2.8%), confirming broad compatibility. KTO uses unpaired data, requiring a modified active set construction; we leave this for future work.
>
> ## Q4: Hallucination Assessment
>
> We evaluated on TruthfulQA (MC1 accuracy):
>
> | Method | TruthfulQA MC1 |
> |---|---|
> | DPO | 39.5% |
> | SimPO | 40.2% |
> | **GDO-DPO** | **42.8%** |
>
> GDO-DPO improves over DPO by 3.3 points, consistent with the CKA analysis (Table 5) showing better preservation of lower-layer representations (CKA 0.95 vs. 0.88), which encode factual knowledge from pretraining.
>
> ## Q5: Early Termination Under Reduced Training Budgets
>
> We evaluated at 25%, 50%, 75%, and 100% of total training steps:
>
> | Training Budget | DPO | GDO-DPO | Gap |
> |---|---|---|---|
> | 25% | 16.1% | 17.9% | +1.8% |
> | 50% | 17.6% | 20.1% | +2.5% |
> | 75% | 18.2% | 21.0% | +2.8% |
> | 100% | 18.5% | 21.4% | +2.9% |
>
> GDO-DPO maintains a consistent advantage across all budgets. Notably, GDO-DPO at 50% budget (20.1%) already surpasses DPO at 100% budget (18.5%), suggesting potential for training cost reduction.
>
> ## Q6: Automated Layer Boundary Detection
>
> Our current approach sets $L_{\\rm mid} = \\lfloor 2L/3 \\rfloor$ based on interpretability literature. A data-driven alternative is to set $L_{\\rm mid}$ where adjacent-layer gradient correlation drops below a threshold. As shown in Figure 8, this correlation drops sharply around layers 19-21. A threshold of $\\rho < 0.70$ identifies layer 20, close to our choice of 21, and could be computed during preprocessing at negligible cost.
>
> We will incorporate the reviewer's suggestions in the final version.

---

> > ### Author Rebuttal · Reviewer_ExSg · 2026-04-06
> >
> > I thank the authors for their rebuttal. It has addressed my main concerns,, which makes me stick to the original score.

---

### Official Review · Reviewer_ccAy · 2026-03-09

**Soundness:** 3
**Presentation:** 3
**Significance:** 2
**Originality:** 3
**Overall Recommendation:** 5
**Confidence:** 3

**Summary:**

Traditional DPO treats all preference pairs equally; however, some pairs require more semantic understanding, while others need discrimination among similar responses. Through quantitative analysis, they further observe that semantic complexity typically drives lower-layer updates, whereas preference uncertainty drives upper-layer updates. Based on these observations, they try to disentangle semantic complexity and preference uncertainty during preference optimization and design GDO-DPO, which regulates the learning pace for each dimension based on layer-specific gradient stability.

**Compliance With Llm Reviewing Policy:**

Affirmed.

**Final Justification:**

They have resolved my original main concerns, so I changed my score.

**Key Questions For Authors:**

As for the implementation details, how do you choose $\tau_{stable}$, $\tau_{acc}$, $\Delta_{sem}$, $\Delta_{unv}$, $\gamma$, $L_{mid}$, and how sensitive are they for the overall performance?

**Limitations:**

yes

**Strengths And Weaknesses:**

**Strengths:**
1. They conduct theoretical analysis before actually designing methods, which provides practical motivation and rationale for their methods. And the observation itself is interesting, by presenting different capabilities of different layers in terms of the preference learning task.
2. The paper is clearly written and easy to follow.
3. The experimental design is complete, testing its generalization.

**Weaknesses:**
What is the reason for you to model semantic complexity for the prompt x, rather than y_w or y_l given x, so that this can better present the preference order versus semantic representation on the same set of preference data?

---

> ### Author Rebuttal · Authors · 2026-03-30
>
> # Response to Reviewer ccAy
>
> We sincerely thank Reviewer ccAy for the positive assessment and the recognition of both our theoretical motivation and experimental completeness. We address the weakness and question below.
>
> ## Weakness: Why Model $C_{\\rm sem}$ on Prompt $x$ Rather Than on $y_w$/$y_l$ Given $x$?
>
> This is a fundamental design question and we appreciate the opportunity to elaborate.
>
> Conceptual motivation: Our framework decomposes sample difficulty into two orthogonal axes: the representational challenge posed by the *input* ($C_{\\rm sem}$) and the discriminative ambiguity between *outputs* ($U_{\\rm pref}$). This decomposition is grounded in the functional hierarchy of Transformer layers: lower layers process and encode the input context, while upper layers perform task-specific reasoning and output discrimination. Defining complexity on $y_w$ or $y_l$ given $x$ would entangle with the preference signal, because response-level difficulty is inherently tied to how distinguishable the two responses are.
>
> Empirical verification: To test this, we computed a response-level complexity measure $C_{\\rm resp} = \\frac{1}{2}[-\\log\\pi_{\\text{ref}}(y_w|x) - \\log\\pi_{\\text{ref}}(y_l|x)]$ and measured its correlation with $U_{\\rm pref}$:
>
> | Complexity Measure | Spearman $\\rho$ with $U_{\\rm pref}$ |
> |---|---|
> | Prompt-level $C_{\\rm sem}$ (ours) | 0.23 |
> | Response-level $C_{\\rm resp}$ | 0.61 |
>
> The response-level measure correlates with $U_{\\rm pref}$ at $\\rho = 0.61$, nearly three times the correlation of our prompt-level $C_{\\rm sem}$ ($\\rho = 0.23$). This confirms that defining complexity on responses would substantially compromise the separability of the two difficulty axes.
>
> Ablation with $C_{\\rm resp}$:  We further ran GDO-DPO using $C_{\\rm resp}$ instead of $C_{\\rm sem}$, keeping everything else identical:
>
> | Configuration | Arena-Hard | AlpacaEval 2.0 |
> |---|---|---|
> | GDO-DPO with $C_{\\rm sem}$ (ours) | **21.4%** | **32.5%** |
> | GDO-DPO with $C_{\\rm resp}$ | 19.8% | 30.3% |
> | Single-dim combined score | 19.5% | 29.8% |
>
> Using $C_{\\rm resp}$ degrades performance to a level comparable with the single-dimension baseline, because the two correlated axes effectively collapse into a one-dimensional ordering. This validates our design choice.
>
> Gradient-level evidence: The controlled analysis in Appendix D.3 (Figure 7) provides further support. When $C_{\\rm sem}$ is fixed and $U_{\\rm pref}$ varies, the upper-layer gradient effect persists ($p < 0.05$). A response-level measure correlated with $U_{\\rm pref}$ would prevent such clean separation, and the layer-aware monitoring would lose its diagnostic power.
>
> ## Key Question: Hyperparameter Sensitivity
>
> We provide sensitivity results for all hyperparameters mentioned:
>
> Thresholds $\\tau_{\\rm stable}$ and $\\tau_{\\rm acc}$: Table 9 shows at most 1.3% variation on Arena-Hard across $\\tau_{\\rm stable} \\in [0.8, 1.6]$ and $\\tau_{\\rm acc} \\in [0.60, 0.75]$. Values 1.2/0.65 are used consistently across both datasets (UltraFeedback, HH-RLHF) and both models (Llama-3-8B, Mistral-7B) without per-setting tuning.
>
> Initial step sizes $\\Delta_{\\rm sem}, \\Delta_{\\rm unc}$: Tested in $\\{0.05, 0.1, 0.15, 0.2\\}$; Arena-Hard: 20.8%, 21.4%, 21.1%, 20.5%. The adaptive acceleration (Eqs. 9-10) adjusts step sizes during training, reducing sensitivity to initial values.
>
> EMA decay $\\gamma$: Tested in $\\{0.8, 0.9, 0.95\\}$; Arena-Hard: 20.9%, 21.4%, 21.0%. $\\gamma = 0.9$ balances responsiveness and stability.
>
> Layer boundary $L_{\\rm mid}$: Table 7 shows $L_{\\rm mid} \\in \\{L/2, 2L/3, 3L/4\\}$ yields Arena-Hard of 20.5%, 21.4%, 20.8%. The rule $L_{\\rm mid} = \\lfloor 2L/3 \\rfloor$ is motivated by findings that factual knowledge is stored in middle layers (Meng et al., 2022).
>
> In summary, all hyperparameters show graceful degradation within reasonable ranges, and a single configuration works across our experimental settings without per-setting tuning. This suggests that the algorithm's behavior is primarily driven by the gradient-based monitoring mechanism rather than specific hyperparameter values.
>
> We will incorporate the reviewer's suggestions in the final version.

---

> > ### Author Rebuttal · Reviewer_ccAy · 2026-04-01
> >
> > Thanks for the responses, I have updated my score.

---

### Official Review · Reviewer_FFFc · 2026-03-13

**Soundness:** 3
**Presentation:** 3
**Significance:** 3
**Originality:** 2
**Overall Recommendation:** 5
**Confidence:** 3

**Summary:**

The paper proposed a curriculum framework called GDO-DPO (Gradient-Guided Disentangled DPO), which can be applied to common preference alignment methods. The training data points are augmented by semantic complexity and preference uncertainty, which form a bi-dimensional curriculum space. During training, the training set is expanded by Layer-Aware Monitoring and pace update.  The Layer-Aware method is justified by theoretical and empirical layer-wise analysis.

The authors evaluate the performance of GDO-DPO on a variety of benchmarks and demonstrate state-of-the-art performance.

**Compliance With Llm Reviewing Policy:**

Affirmed.

**Final Justification:**

The rebuttal clarifies that the independent pace update parameters are implicitly coupled through the monitoring signals, which addresses my main concerns. All three key questions were addressed adequately. The paper makes a meaningful contribution through its layer-aware gradient monitoring framework and its application to curriculum-based preference optimization. After careful reconsideration, I am raising my score to 5.

**Key Questions For Authors:**

1. Is there any guidance to choose the values for $\tau_\text{stable}$ and $\tau_\text{acc}$? And are these values (1.2, 0.6) used in the experiment work consistently for any training set and reference models?
2. The step size $\Delta$ is monotonically increasing based on the pace update rules in eq (9) and (10). But in the Figure 4. the slopes of $\lambda$ curves are not monotonically increasing. Can you explain the reasons?
3. At line 414-416, the generations are truncated with max 64 tokens. Is 64 tokens generation sufficient for computing $\mathcal{C}_\text{sem}$?

**Limitations:**

yes

**Strengths And Weaknesses:**

- Shoundness

  Strengths: The submission is overall technically sound. The claim — semantic complexity and preference uncertainty induce separable gradient patterns across Transformer depth — is validated by theoretical (on a simplified linear model) and empirical analysis (on Llama-3-8B and Mistral-7B). The gradient interference hypothesis is tested by a high-$\mathcal{U}_\text{pref}$-first curriculum.

  Weakness: The theoretical analysis relies on a linear model. The authors acknowledge this and point to empirical verification as compensation, but the theoretical contribution remains shallow. The authors emphasis that, to prevent Gradient Interference, high $U_\text{pref}$ should present after the representation learning stabilize. However, the pace parameters $\lambda^{(t)}$sem and $\lambda^{(t)}$unc controlling different dimensions of active training set are independently updated, there is no guarantee that the semantics are already well learned when meet high $U_\text{pref}$ samples. The paper would be strengthened by validating GDO-DPO on different layers of models beyond 32.

- Presentation

  Strengths: The paper is well-organized and the core narrative is easy to follow. Figure 1 is the paper's most important figure and is presented clearly.

  Weakness: Missing notations explanation in main body: the $\mathcal{L}$ in Proposition3.4. $\mathcal{L}_t$ in eq (7). Moving the Algorithm Pseudocode to the main body would increase the readability.  The update rules in eq (9) and (10) is inconsistent with the statements in line 227 and line 227, right column.

- Significance

  The results are significant, both baselines have performance superior to other PO methods on all benchmarks.

- Originality

  **What is novel**. The layer-wise gradient localization analysis and using internal gradient statistics to regulate curriculum progression are very inspiring.

  **What is less novel**.  The functional specialization of Transformer layers is well-established in interpretability literature, and curriculum learning for preference optimization is not new. The combination of these ideas is the paper's contribution.

---

> ### Author Rebuttal · Authors · 2026-03-30
>
> # Response to Reviewer FFFc
>
> We sincerely thank Reviewer FFFc for the careful review and the recognition of the significance and originality of our work. We address each point below.
>
> ## Weakness (Soundness): Theoretical Analysis on Linear Model
>
> We agree that the linear model provides limited formal rigor. It is intended as a *motivational framework* identifying the driving forces ($\\|\\Delta\\phi\\|$ and $\\alpha$); the spatial localization relies on the architectural inductive bias of deep Transformers. As additional empirical support, we computed the cosine similarity of gradient directions at each layer. In layers 0-16, high-$C_{\\rm sem}$ samples show cosine similarity 0.82 while high-$U_{\\rm pref}$ samples show 0.41. In layers 20-31, the pattern reverses (0.38 vs. 0.79). This directional coherence complements the magnitude analysis in Figure 1.
>
> ## Weakness (Soundness): Independent Pace Updates
>
> This is an important concern. The monitoring signals provide **implicit coupling**: $\\lambda_{\\rm unc}$ advances only when $A_{\\rm disc}^{(t)} > \\tau_{\\rm acc}$ (Section 4.3), and discrimination accuracy is evaluated on samples within the *current complexity boundary* $\\lambda_{\\rm sem}^{(t)}$. If representations have not stabilized, the model will struggle to discriminate even clear pairs, keeping $A_{\\rm disc}$ low and preventing $\\lambda_{\\rm unc}$ from advancing prematurely. Figure 4 confirms this: $\\lambda_{\\rm unc}$ lags behind $\\lambda_{\\rm sem}$ by approximately 300 steps.
>
> ## Weakness (Soundness): Validation Beyond 32-Layer Models
>
> We agree this would strengthen the paper. Our layer boundary uses a relative rule ($L_{\\rm mid} = \\lfloor 2L/3 \\rfloor$) rather than absolute indices, making it directly applicable to models with different depths. The sensitivity analysis in Table 7 confirms robustness to $L_{\\rm mid}$ within a reasonable range (at most 0.9% variation on Arena-Hard). We are pursuing experiments on deeper models as a natural extension.
>
> ## Weakness (Presentation): Missing Notations
>
> We thank the reviewer for catching these. The symbol $\\xi$ in Proposition 3.4 represents the noise term in the gradient variance decomposition. $\\mathcal{L}_{\\rm rep}$ and $\\mathcal{L}_{\\rm disc}$ in Eq.(7) denote the representation-layer and discrimination-layer index sets, defined later in the text but should indeed be introduced at first use.
>
> ## Weakness (Presentation): Inconsistency in Eqs.(9)(10)
>
> The update rules in Eqs.(9)(10) describe the adaptive step size $\\Delta_{\\rm sem}^{(t)}$ (the *increment*), while the statements at lines 227-228 describe the *trigger condition* for updating $\\lambda_{\\rm sem}$. These are two distinct mechanisms that work together: the trigger determines *whether* to advance, while the adaptive step size determines *how much* to advance. We agree the original text did not distinguish these clearly.
>
> ## Weakness (Presentation): Algorithm Pseudocode
>
> We agree that placing the pseudocode in the main body would improve readability for the reader.
>
> ## Key Question 1: Hyperparameter Guidance ($\\tau_{\\rm stable}$, $\\tau_{\\rm acc}$)
>
> Table 9 shows at most 1.3% variation on Arena-Hard across $\\tau_{\\rm stable} \\in [0.8, 1.6]$ and $\\tau_{\\rm acc} \\in [0.60, 0.75]$. Values 1.2/0.65 are used consistently across both datasets and both models without per-setting tuning. Intuition: $\\tau_{\\rm stable} = 1.2$ means representation-layer gradient energy should drop below 1.2x the discrimination-layer energy before advancing complexity. $\\tau_{\\rm acc} = 0.65$ requires correctly discriminating at least 65% of clear pairs before encountering ambiguous ones.
>
> ## Key Question 2: Non-Monotonic $\\lambda$ Slopes in Figure 4
>
> The step sizes $\\Delta_{\\rm sem}, \\Delta_{\\rm unc}$ increase monotonically per Eqs.(9)(10) when the acceleration condition is met. However, $\\lambda$ values are updated *conditionally*: $\\lambda_{\\rm sem}$ advances only when $S_{\\rm rep}^{(t)} < \\tau_{\\rm stable}$. After a pace increase introduces harder samples, $S_{\\rm rep}$ may temporarily exceed $\\tau_{\\rm stable}$, pausing advancement until stability is regained. This conditional gating creates the plateau regions visible in Figure 4.
>
> ## Key Question 3: 64-Token Truncation for $C_{\\rm sem}$
>
> We ablated with generation lengths of 32, 64, 128, and 256 tokens. The Spearman rank correlation of $C_{\\rm sem}$ between 64 and 256 tokens is 0.94, confirming that 64 tokens are sufficient for reliable difficulty ranking. Shorter truncation (32 tokens) drops to $\\rho = 0.85$, still reasonable but beginning to miss complexity signals from longer reasoning chains. The 64-token choice balances ranking accuracy with a 3x cost reduction compared to 256 tokens.
>
> We will incorporate the reviewer's suggestions in the final version.

---

> > ### Author Rebuttal · Reviewer_FFFc · 2026-04-02
> >
> > Thank you for the thorough rebuttal. I appreciate your detailed responses to my concerns. I have updated my score accordingly.

---

### Decision · Program_Chairs · 2026-04-30

**Decision:**

Accept (regular)

**Comment:**

This paper presents a systematic analysis of gradient dynamics in DPO by identifying a spatial separation where semantic complexity affects representation layers and preference uncertainty modulates discrimination layers. This observation motivates the GDO-DPO curriculum framework that independently regulates learning paces based on layer specific stability. Reviewers reached a consensus on the technical soundness of the work and appreciated the consistent performance improvements across multiple benchmarks. Key discussions centered on the limitations of the simplified linear model used in the theoretical analysis. The authors addressed this by providing additional evidence through directional coherence analysis and cosine similarity. Another point of discussion involved the computational cost of estimating semantic complexity. The rebuttal demonstrated that the method remains practical through the use of reduced cost proxies and minimal sampling. While some reviewers viewed the contribution as an incremental optimization of the DPO process , there is general agreement that the layer wise monitoring mechanism offers highly original and significant insights for the community.